# Iron ions regulate antifungal HSAF biosynthesis in *Lysobacter enzymogenes* by manipulating the DNA-binding affinity of the ferric uptake regulator (Fur)

Bao Tang,[1,2] Bo Wang,[1] Zhizhou Xu,[1,3] Rouxian Hou,[1,3] Min Zhang,[1] Xian Chen,[1] Youzhou Liu,[1] Fengquan Liu[1,4]

**ABSTRACT**    Heat-stable antifungal factor (HSAF), produced by *Lysobacter enzymogenes* OH11, is regarded as a potential biological pesticide due to its broad-spectrum antifungal activity and novel mode of action. However, the current production of HSAF is low and cannot meet the requirements for large-scale production. Herein, we discovered that iron ions greatly promoted HSAF production, and the ferric uptake regulator (Fur) was involved in this regulatory process. Fur was also found to participate in the regulation of iron homeostasis in OH11 via the classic inhibition mechanism of *Holo*-Fur. Furthermore, Fur was collectively observed to directly bind to the promoter of the HSAF biosynthesis gene, and its DNA-binding affinity was attenuated by the addition of iron ions *in vitro* and *in vivo*. Its regulatory mechanism followed the uncommon inhibition mechanism of *Apo*-Fur. In summary, Fur exhibited a bidirectional regulatory mechanism in OH11. This study reveals a novel regulatory mechanism whereby Fur upregulates the biosynthesis of secondary metabolites. These findings contribute to the improvement of HSAF production and may guide its development into biological pesticides.

**IMPORTANCE**    HSAF possesses potent and broad antifungal activity with a novel mode of action. The HSAF yield is critical for fermentation production. In this study, iron ions were found to increase HSAF production, and the specific mechanism was elaborated. These results provide theoretical support for genetic transformation to improve HSAF yield, supporting its development into biological pesticides.

**KEYWORDS**    *Lysobacter enzymogenes*, biological pesticides, ferric uptake regulator, heat-stable antifungal factor

*L*ysobacter spp., belonging to the *Xanthomonadaceae* family of *Gammaproteobacteria,* has been recently demonstrated as a potential biological control agent against bacterial and fungal diseases (1). The genus not only secretes a large amount of extracellular hydrolases (α- proteolytic enzyme, β-1,3-glucanase, chitinase, etc.) but also produces many bioactive natural products (NPs) (2–4). Polycyclic tetramate macrolactams have been the most extensively studied NPs, particularly the heat-stable antifungal factor (HSAF) from *Lysobacter enzymogenes* strains OH11 and C3, which shows a broad-spectrum antifungal activity with a distinct mode of action (5–7). Thus, HSAF has great potential use as a biological pesticide for sustainable and safe agricultural production.

The biosynthetic pathway and regulatory mechanisms of HSAF are fairly complex. First, the synthesis of HSAF is mainly implemented in *L. enzymogenes* by an operon including more than 10 genes. Its biosynthesis pathway is as follows. Polyketide synthase is responsible for synthesizing two polyketide chains and is assembled with one ornithine through non-ribosomal peptide synthetase to finally form the scaffold

Address correspondence to Fengquan Liu, fqliu20011@sina.com.

Bao Tang and Bo Wang contributed equally to this article. Author order was determined on the basis of contribution.

The authors declare no conflict of interest.

structure of HSAF (8, 9). This mode is unique among the biosynthesis pathways of most known NPs. Furthermore, the regulation of HSAF production mainly occurs at the level of gene expression. Up to now, a total of six transcription regulators regulating HSAF synthesis have been identified, which can be divided into two categories: five positive regulators [PilR (10), Clp (11), LysR (12), LarR (13), and CdgL (14)] and one negative regulator [LetR (15)]. In addition, the fermentation medium and conditions have been optimized to increase the HSAF production (16, 17). Despite these efforts, current HSAF production (440.26 ± 16.14 mg/L) is far from the requirements for large-scale fermentation production (g/L). Therefore, more mechanisms need to be explored to further improve HSAF production for the development of biological pesticides.

Previous research has shown that the antagonistic activity of *L. enzymogenes* is coordinated with the availability of nutrients (18). In nutrient-rich medium, including lysogeny broth (LB) or tryptic soy broth (TSB), very little HSAF is synthesized, compared with nutrient-starved medium (10% LB or 10% TSB) (5). However, it is still unknown which component of the medium has a significant impact on the synthesis of HSAF. The main reason is that the research of HSAF is carried out in natural medium, the composition of which is not clear. Thus, the identification of the medium component that regulates HSAF production is of significant interest.

In this study, iron ions were found to significantly increase HSAF production in *L. enzymogenes* OH11. Then, a ferric uptake regulator (Fur) was identified to be involved in this regulatory process by transcriptomics analysis and deletion mutation analysis. Moreover, the regulatory mechanism of Fur was elucidated. In a word, we report the function of Fur in regulating secondary metabolism in *Lysobacter* for the first time, which will facilitate increased HSAF production and its development into a biological pesticide.

## RESULTS

### Iron ions promote the production of HSAF by *L. enzymogenes* OH11

Previously, we have studied the effects of carbon sources, nitrogen sources, and inorganic salts on HSAF production (16). In order to further increase the production of HSAF, the effects of different metal ions on the fermentation were investigated in the chemically defined medium 502 (CDM502), as shown in Fig. 1. Compared with the control, the addition of different types of metal ions promoted cell growth to some extent yet had an inconsistent impact on HSAF production (Fig. 1A). The HSAF biosynthesis was inhibited by some divalent cations such as $Mn^{2+}$, $Cu^{2+}$, and $Zn^{2+}$. Instead, the presence of $Fe^{2+}$, $Fe^{3+}$, $Mg^{2+}$, and $Ca^{2+}$—particularly $Fe^{2+}$ and $Fe^{3+}$—had a promotive effect, including a five- to sixfold improvement in HSAF production. Considering the unstable nature of $Fe^{2+}$, it was rapidly oxidized under aerobic and neutral conditions (19), so $Fe^{3+}$ was selected as the suitable metal ion for subsequent experiments. Furthermore, the concentrations of $Fe^{3+}$ were optimized. With increased concentration, HSAF production also increased gradually with no significant change in cell growth (Fig. 1B). The highest production of 273.83 ± 4.57 mg/L was achieved at an $Fe^{3+}$ dose of 16 mg/L.

### Transcriptomics analysis of differentially expressed genes in response to iron ions in *L. enzymogenes* OH11

To elucidate the iron-response mechanism in HSAF biosynthesis, the whole transcriptomic profile changes of OH11 were investigated using RNA-seq. Compared with the transcripts without iron, a total of 554 differentially expressed genes (DEGs) were identified in the iron-treated samples, of which 300 genes were upregulated and 254 genes were downregulated (Fig. 2A). The validation results of reverse transcription quantitative PCR (RT-qPCR) were consistent with that of the RNA-seq data (Fig. S1; Table S1). The three most represented terms of Cluster of Orthologous Groups were "energy production and conversion (C)," "general function prediction only (R)," and "inorganic ion transport and metabolism (*P*)." In addition, the terms "transcription (K)" and "secondary metabolite biosynthesis, transport, and catabolism (Q)" also accounted for a large

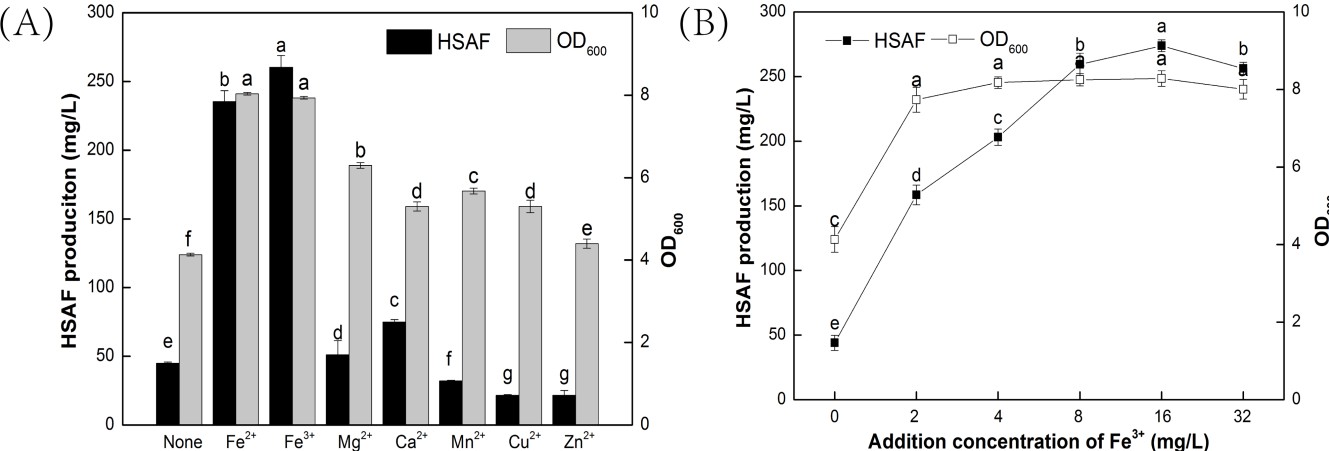

**FIG 1** Effects of metal ion types (A) and concentrations (B) on the fermentation. Significant differences calculated by Tukey's least significant difference, $P < 0.05$. Small letters correspond to significance values of different treatment.

proportion. Within transcription (K), the most common gene was Rrf2 family transcriptional regulator (−3.87-fold), followed by ferric iron uptake transcriptional regulator (−2.36-fold). Within the secondary metabolite biosynthesis, transport, and catabolism (Q), the most common gene was involved in the biosynthesis of HSAF (1.29-fold) (Fig. 2B). These observations imply that iron may regulate the synthesis of secondary metabolites by affecting gene transcription levels. Thus, we searched for genes related to iron metabolism in these terms. A total of six genes were screened that may be involved in the regulation of HSAF biosynthesis, including (2Fe-2S)-binding protein (4.88-fold, Bp, protein number in OH11: OH11GL003062), Bacterioferritin (4.49-fold, Bfr, protein number in OH11: OH11GL002596), Ferrous iron transport protein A (−3.87-fold, FitpA, protein number in OH11: OH11GL005044), Rrf2 family transcriptional regulator (−3.87-fold, Rrf2, protein number in OH11: OH11GL003187), Ferric iron uptake transcriptional regulator (−2.36-fold, Fur, OH11GL004785), and putative DNA-binding ferritin-like protein (1.29-fold, Dps, OH11GL000618).

## Identification of iron-related genes involved in regulating HSAF production

To examine the roles of the screened genes in cell growth and HSAF biosynthesis, six gene (*bp*, *bfr*, *fitp*A, *rrf2*, *fur*, and *dps*) deletion mutants were constructed through homologous recombination and cultured in the CDM502 with different concentrations of

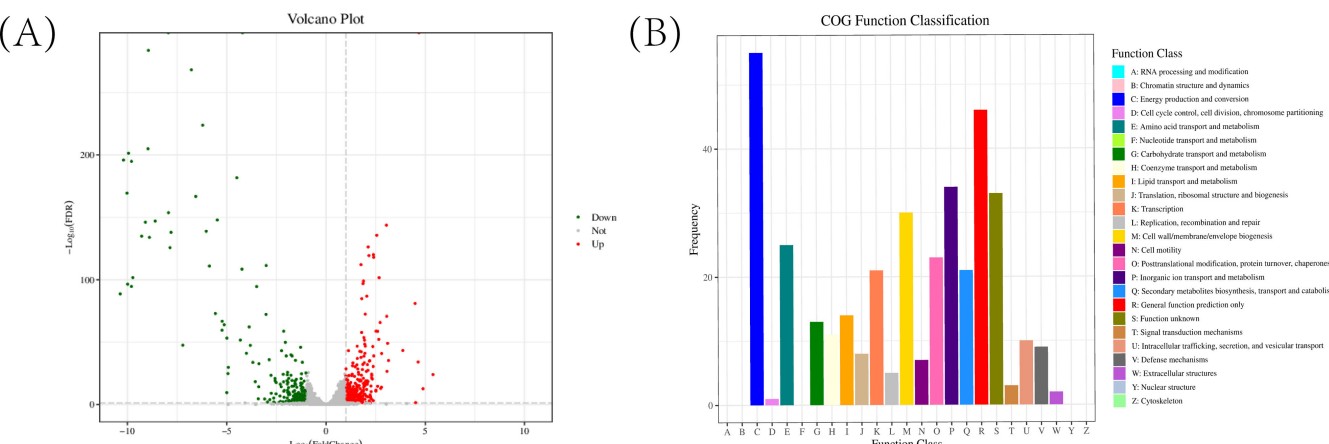

**FIG 2** Gene Ontology classification map of DEGs in response to iron ions in *L. enzymogenes* OH11. (A) Volcano plot of DEGs. Significantly differentially expressed genes are shown as a red (up) or green (down) dot. No significantly expressed genes are shown as a gray dot. (B) Summary of COG annotation of the DEGs.

$Fe^{3+}$. Except for the *fur* gene, the deletion of other genes did not induce changes in cell growth and HSAF production (Fig. S2). Compared *with* wild-type (*WT*), the cell growth of Δ*fur* was lower but not significantly (Fig. 3A). However, there was a significant difference in HSAF production between Δ*fur* and *WT* under different concentrations of iron ions (Fig. 3B). At low concentrations of iron ion (≤4 mg/L), the HSAF production of Δ*fur* was significantly higher than that of *WT*. Moreover, this significance became increasingly obvious with the decrease in iron ion concentration. In particular, the production of HSAF reached 161.34 ± 5.87 mg/L when the medium was completely iron deficient, which was 3.58-fold higher than that of *WT* (45.06 ± 5.94 mg/L). When the concentrations of $Fe^{3+}$ exceeded 8 mg/L, there was no difference in HSAF production between Δ*fur* and *WT*. Furthermore, a complemented strain [Δ*fur* (*fur*)] and an overexpression strain [*WT* (*fur*)] with the recombinant plasmid pBBR-*fur* were constructed and cultured. As shown in Fig. 3, the growth ability and HSAF production of Δ*fur* (*fur*) were restored to a certain extent of *WT*, but not completely. The *WT* (*fur*) grew significantly faster than *WT*, Δ*fur*, and Δ*fur* (*fur*) in the high-iron medium (≥8 mg/L). It is likely that the enhanced expression of *fur* increased the metabolism of iron in the cell, which was beneficial to cell growth. On the contrary, the ability to produce HSAF was reduced greatly compared with other strains.

## Fur is a highly conserved global transcription factor that contains one DNA-binding domain and two metal-binding sites

To further explain the mechanism by which Fur regulates HSAF production, bioinformatics analysis of Fur was performed. The Fur protein was composed of 133 amino acid residues with a molecular weight (MW) of 15.4 kDa, and its isoelectric point (pI) was 5.83. Comparison of the amino acid sequences indicated that Fur proteins in different bacteria are highly conserved and homologous. In particular, the degrees of identity reached 95.41%–92.48% within the same genus of *Lysobacter* (Fig. 4A). The prediction of protein secondary structure showed that Fur in OH11 contained five α-spiral and five β-collapse domains. Of these, β3 before folding possessed a typical HHDH ion-binding region, the function of which might be related to the formation of dimer structure. The three-dimensional structure model revealed that Fur in OH11 (gray) was also highly conserved with Fur in *Pseudomonas aeruginosa* (cyan), for which one DNA-binding domain and two metal-binding sites have been reported: one is responsible for regulation and the other for structural stabilization (Fig. 4B) (20).

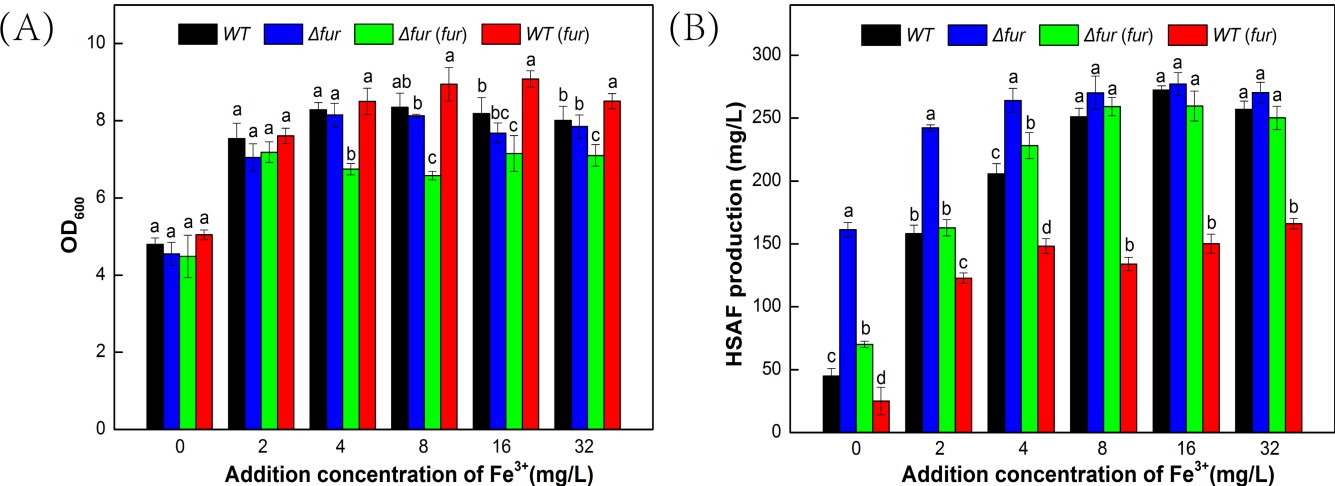

**FIG 3** Cell growth (A) and HSAF production (B) of strains *WT*, Δ*fur*, Δ*fur* (*fur*), and *WT* (*fur*) in culture with different concentrations of $Fe^{3+}$. Significant differences calculated by Tukey's least significant difference, $P < 0.05$. Small letters correspond to significance values of different treatmentS.

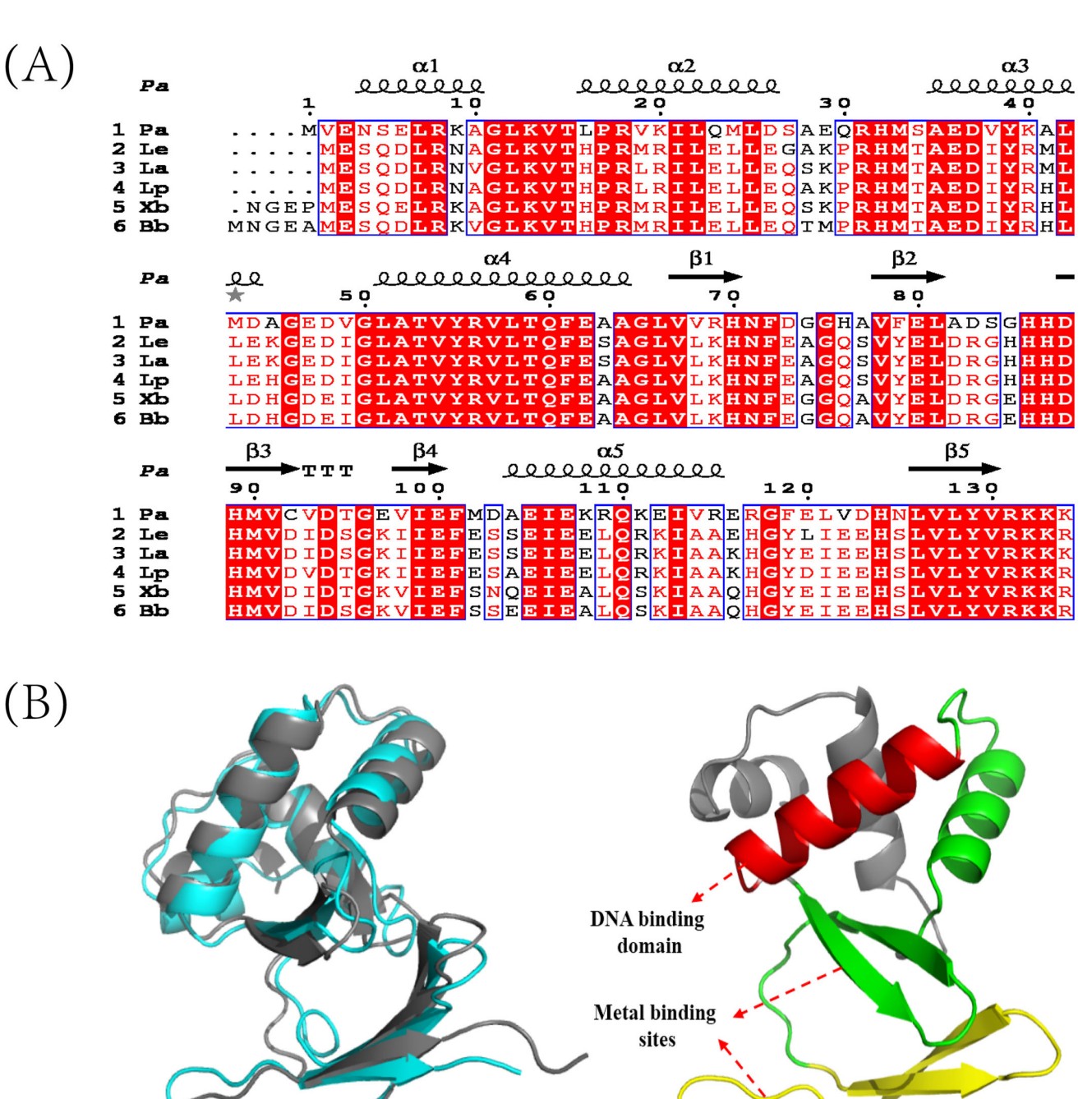

**FIG 4** Protein sequence and structure alignment. (A) Alignments were performed with ClustalW based on identical residues. The alignments were graphically rendered, and the structure of the Pa structure as a reference (PDB ID: 1MZB) was superimposed using ESPript. Structural elements, such as α-helices, turns (T), and β-strands (arrows), are indicated. Amino acid positions are denoted on the top of the sequence. Conserved residues are highlighted by red. (B) Three-dimensional structure and structure alignment of Fur(Pa) and Fur(Le) by the TM-align database. Pa, *Pseudomonas aeruginosa*; Le, *Lysobacter enzymogenes*; La, *Lysobacter antibioticus*; Lp, *Lysobacter panacisoli*; Xb, *Xanthomonadaceae bacterium*; Bb, *Burkholderiales bacterium*.

## Effect of the deletion mutation of *fur* on iron metabolism

In most bacteria, the transcription factor Fur has a major role in iron acquisition, transportation, and storage in order to adapt to an iron-deficient environment (21). To test whether the *fur* deletion mutation affected iron ion homeostasis, the intracellular iron concentration of strains *WT*, Δ*fur*, and Δ*fur* (*fur*) was measured by inductively coupled plasma mass spectrometry (ICP-MS). Under iron-deficient conditions, the intracellular iron concentration in Δ*fur* was 9.49 ± 0.28 µg/g cell, a 7.95% reduction compared to that in *WT* (Fig. 5). This indicates that the mutation of *fur* caused the reduction of intracellular iron, which may affect the growth of the mutant under iron deficiency. Under iron-rich conditions, the intracellular iron concentration in Δ*fur* was 59.57 ± 1.24 µg/g cell, which represents a 23.28% increase compared to that in *WT*. The *fur* complement made the intracellular iron content restore the *WT* to a certain extent.

Since Fur plays a negative regulatory role in the transport system of iron ions (22, 23), it was speculated that the mutation of *fur* leads to the enhancement of the intracellular active transport capacity of iron ions. To further verify this result, the transcription levels of genes related to iron transport in OH11 were measured by RT-qPCR. The genes of *OH11GL001789*, *OH11GL002528*, and *OH11GL003262* were selected according to a previous report (24) and homologous comparison, encoding TonB transporter, ABC transporter ATP-binding protein, and TonB-dependent receptor, respectively. As expected, the transcription levels of *OH11GL001789* and *OH11GL003262* were slightly lower (0.85- and 0.92-fold, respectively), whereas those of *OH11GL002528* were slightly higher (1.15-fold) in Δ*fur* relative to *WT* under iron-limited conditions. When iron ion was

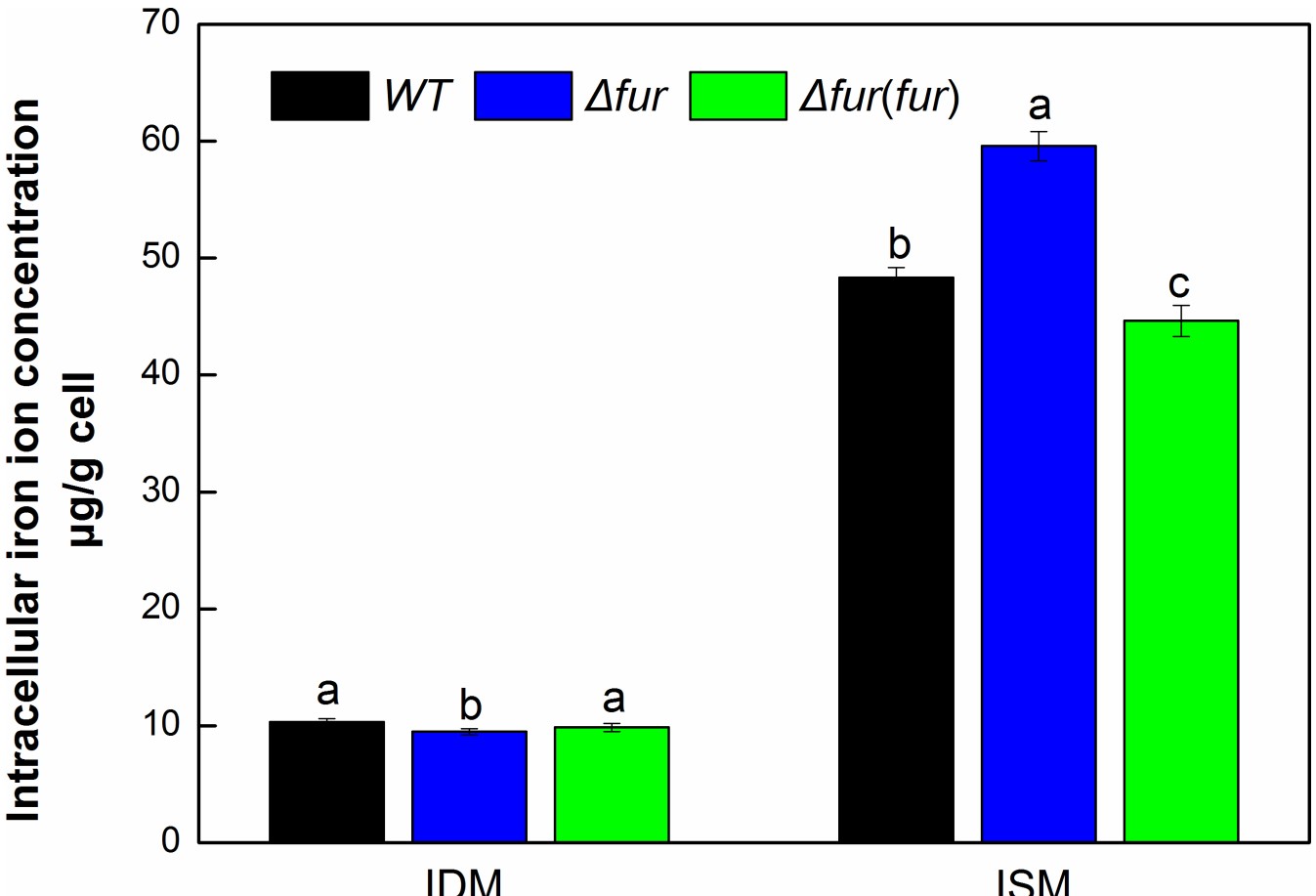

**FIG 5** Intracellular iron concentration of strains *WT*, Δ*fur*, and Δ*fur* (*fur*) cultured in the IDM and ISM. Significant differences calculated by Tukey's least significant difference, $P < 0.05$. Small letters correspond to significance values of different treatment. IDM, iron deficiency medium; ISM, iron sufficiency medium.

added to the medium, the transcription levels of iron transport-related genes in Δ*fur* were significantly improved compared to those in *WT*, particularly *OH11GL001789*, which was associated with a 5.61-fold increase (Fig. 6). In brief, the deletion mutation of *fur* enhanced the expression of iron-transport genes, thus increasing the intracellular iron concentration.

## Iron ions inhibit the affinity of Fur to bind to the promoter of the HSAF biosynthesis gene

Fur is a global regulator that controls its targets by directly or indirectly binding to the promoter regions of target genes (22). We therefore wondered whether it interacts with the promoter of the HSAF biosynthesis gene ($P_{HSAF}$). To assess this, electrophoresis mobility shift assay (EMSA) and bacterial one-hybrid assay were simultaneously executed.

As shown in Fig. S3A, $P_{HSAF}$ was amplified with 400 bp of DNA fragments and regarded as the probe in EMSA. The sodium dodecyl sulphate–polyacrylamide gel electrophoresis (SDS-PAGE) analysis of the expressed His6-Fur showed migration at ~22.36 kDa (Fig. S3B), and therefore, it was subjected to *in vitro* binding assays. Once the protein of Fur was added to this reaction, the retarded migration could be observed

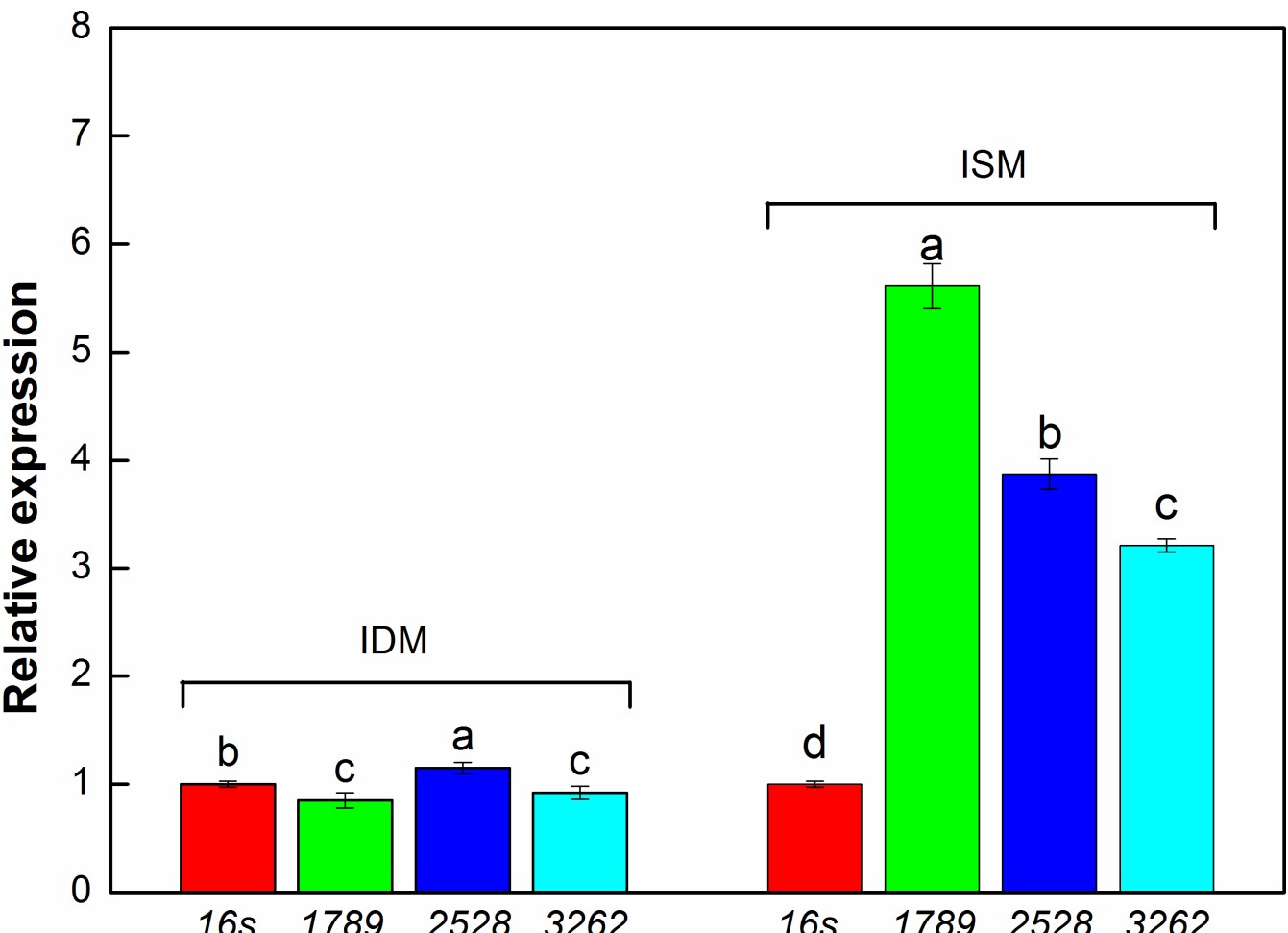

**FIG 6** Transcription levels of iron transport genes in *WT* and Δ*fur* cultured in the IDM and ISM. 16S ribosomal RNA was used as the endogenous control; the expression level in the wild-type strain was assigned a numerical value of 1; and transcript levels were normalized to the 16S RNA level. Experiments were performed three times with similar results. Each column indicates the mean of three biologically independent experiments. Vertical bars represent standard errors. Significant differences were calculated by Tukey's least significant difference, $P < 0.05$. Small letters correspond to significance values of different treatment.

on the PAGE gel in the absence of iron ion. Moreover, the migration bands appeared increasingly stronger, while the free $P_{HSAF}$ bands appeared increasingly weaker as the amount of Fur increased from 0.2 to 0.6 µM (Fig. 7A, lanes 2 through 5). These results indicated that the purified Fur could directly bind $P_{HSAF}$ *in vitro*, which formed a complex, and the binding did not require the addition of iron ion. Although ferric ion ($FeCl_3$) was added in the ISM, it was finally transformed into ferrous ion in the cells to perform its function (24). Additionally, Fur is known to be active in a dimeric form in which each monomer is believed to bind one ferrous ion (25). Thus, $FeSO_4$ was added to the EMSA reaction to investigate the effect of iron ion on their binding. As shown in Fig. 7A and lanes 6 through 7, the complex formation was weakened or even disappeared with the addition of $FeSO_4$, suggesting that iron ion inhibited the DNA-binding ability of Fur. Here, we use the promoter of *OH11GL003062* as a negative control of EMSA analysis. As shown in Fig. 7B, Fur cannot be directly bound to $P_{OH11GL003062}$, which means that the binding between Fur and $P_{HSAF}$ is specific. In addition, a bacterial one-hybrid assay was carried out to further determine the combination of Fur and $P_{HSAF}$ under different iron conditions *in vivo*. As expected, the co-expression strain harboring the plasmids pTRG-*fur* and pBXcmT-$P_{HSAF}$ grew as well as the positive strain on the M9 plate (Fig. 7C), implying a combination of Fur and $P_{HSAF}$. With the addition of $FeSO_4$, the co-expression strain grew more slowly than the positive strain under the same culture conditions, indicating that their interaction was inhibited by iron ion.

These results suggest that Fur can directly interact with $P_{HSAF}$, and the specific interactions were inhibited by iron ion.

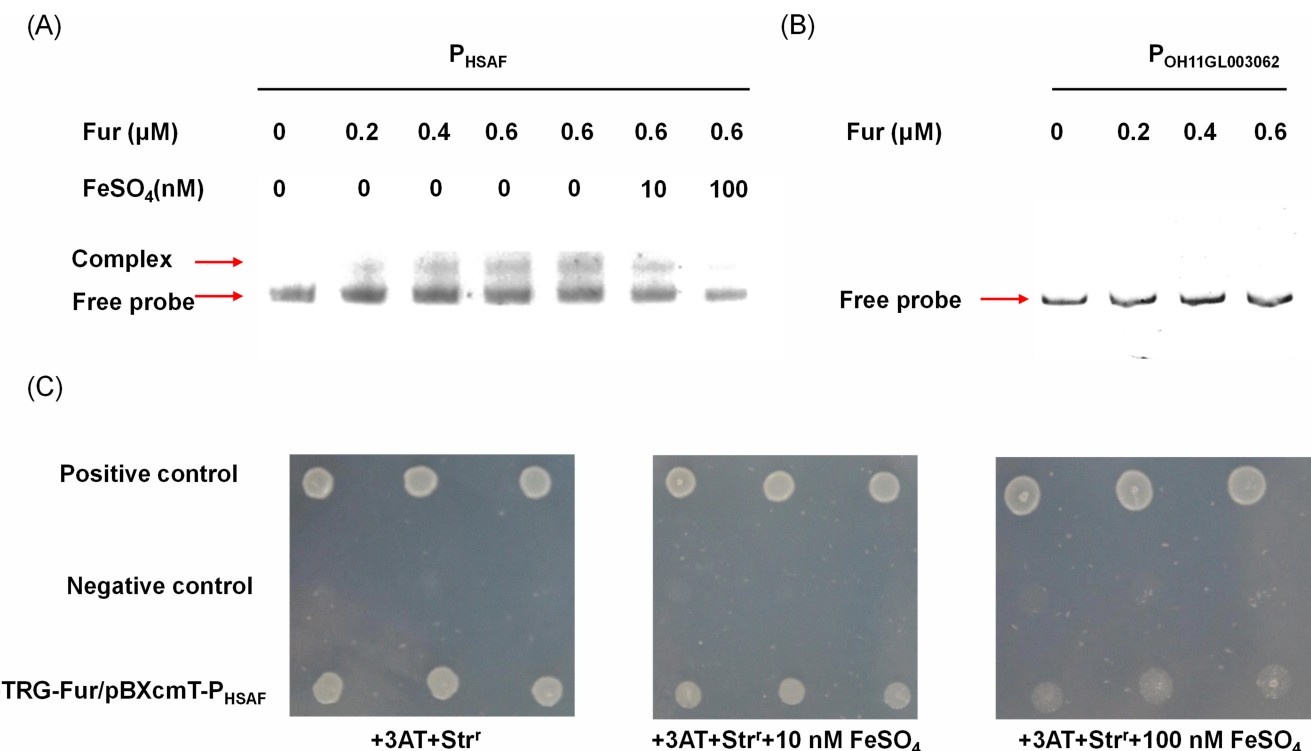

FIG 7 Iron ions inhibit the binding between Fur and $P_{HSAF}$. (A) Fur binds to $P_{HSAF}$ in *vitro*. Gel shift assay showing that Fur directly binds to $P_{HSAF}$, and the formation of the complex could not be weakened by adding 10-nM $FeSO_4$. Different concentrations of Fur were added to reaction mixtures containing 20 ng of probe DNA, and the reaction mixtures were separated on polyacrylamide gels. (B) Gel shift assay showing that Fur does not directly bind the *OH11GL003062* promoter region. Different concentrations of Fur were added to reaction mixtures containing 20 ng of probe DNA, and the reaction mixtures were separated on polyacrylamide gels. (C) The direct physical interaction between Fur and $P_{HSAF}$ was detected in *Escherichia coli* under different iron ions growth condition. Positive control, co-transformant containing pBX-R2031 and pTRG-R3133; negative control, co-transformant containing pBXcmT-$P_{HSAF}$, and empty pTRG; pTFur/pBxcmT-$P_{HSAF}$, co-transformant possessing both pTRG-Fur and pBXcmT-$P_{HSAF}$; −3AT-Str$^r$, 3AT, 3-amino-1,2,4-triazole; +3AT + Str$^r$, M9-based selective medium plate; Str, streptomycin.

## DISCUSSION

Iron ion is an essential nutrient for the survival and growth of most microorganisms (26). As an indispensable co-factor, iron ions (ferrous iron $Fe^{2+}$ and ferric iron $Fe^{3+}$) are involved in crucial biochemical processes in cells, including respiration, central metabolism, nitrogen fixation, DNA biosynthesis, and gene regulation (27, 28). Although iron is the fourth most abundant element in the Earth's crust, the availability of iron is still limited for bacteria due to its poor solubility. Microorganisms have evolved various systems to uptake iron and maintain homeostasis. In this study, the survival of OH11 was not completely dependent on iron ions, though their addition was beneficial to the growth process. Surprisingly, we found that iron ions substantially promoted HSAF production in the CDM502 (Fig. 1). The RNA-seq data showed that 10.53% of genes in OH11 were influenced by the addition of ferric ion, including genes related to basic metabolism, transcriptional regulation, and secondary metabolite synthesis (Fig. 2). Furthermore, the transcription factor of Fur was found to participate in the regulatory process of HSAF biosynthesis through the negative regulation mode under low-iron conditions, but it is not involved in the cell growth of OH11 (Fig. 3). As previously reported, Fur also participated in the regulation of intracellular iron concentration in OH11 (Fig. 5). The sequences and structures of Fur in OH11 were highly conserved and homologous with other bacteria (Fig. 4), hinting that Fur here, similar to other homologous transcription factors, may bind to the promoter of target genes to regulate the biosynthesis of HSAF.

Also of note is that *fur* has been reported as an essential gene in a few bacteria, such as *Pseudomonas stutzeri* (29). Recent studies have described it as a conditional essential gene (30, 31). The reasons why *fur* is important and even an essential gene in some bacteria remain elusive, but multiple factors are likely involved (31). Indeed, obtaining the deletion mutation of *fur* in OH11 was quite difficult, and double-crossover recombinants were only acquired on the sucrose plate with the addition of ferric ion. Therefore, our findings may provide a reference for the knockout of the *fur* gene in other species.

Fur is a global regulator that was first identified in *Escherichia coli* and basically functions as an iron sensor that maintains iron homeostasis in various bacteria by controlling iron transporter-coding genes (32). In addition, Fur regulates a variety of crucial physiological and metabolic pathways in most prokaryotes, including DNA synthesis (33), the tricarboxylic acid cycle (34), swimming motility (35), biofilm formation (22), sensitivity to acid stress (32), bacterial virulence (26), and siderophore biosynthesis (29). To date, the role of Fur in regulating the biosynthesis of secondary metabolites has mainly focused on siderophores. Our results showed that Fur also regulates the agricultural antibiotic HSAF, which represents a new discovery.

At present, Fur has five regulatory mechanisms: *Holo*-Fur repression, *Holo*-Fur activation, *Apo*-Fur repression, *Apo*-Fur activation, and *Holo*-Fur indirect activation. Holo-Fur refers to the dimer that binds iron ions, and Apo-Fur refers to the independent protein that does not bind iron ions (36). Generally, Fur follows the classic repression pattern of *Holo*-Fur in which Fur binds to "Fur-box" in the promoter to restrict the binding of RNA polymerase and represses gene transcription (37). Our study revealed that Fur could directly bind to $P_{HSAF}$, and the addition of iron ions attenuated its DNA-binding affinity. This result indicates that Fur can respond to intracellular iron ions to regulate HSAF biosynthesis, following the repression mechanism of *Apo*-Fur described below. When the intracellular iron concentration is low, *Apo*-Fur can bind to the promoter of the gene to inhibit its transcription. When the intracellular iron concentration is high, Fur usually combines with iron ions to form a dimer, which cannot bind to the promoter (38). *Apo*-Fur regulation rarely occurs but has been experimentally demonstrated in *Helicobacter pylori*, in which *Apo*-Fur inhibits the expression of superoxide dismutase sodB, iron storage molecule pfr, and fatty amidase amiE in a unique form of transcriptional repression under iron-deficient conditions, despite the conservation among bacterial Fur proteins (28, 39).

In summary, Fur exhibits a bidirectional regulatory mechanism in OH11 whereby the regulation of iron uptake follows the classic inhibitory mechanism of *Holo*-Fur, but the regulation of HSAF follows the uncommon inhibitory mechanism of *Apo*-Fur. The proposed model could be described as follows (Fig. 8): under iron deficiency , Fur is induced to be highly expressed, yet no Fur-$Fe^{2+}$ complexes are available to act as repressors. In turn, the iron uptake process is stimulated and helps bacteria to obtain more iron from the external environment. Additionally, native Fur can bind to the $P_{HSAF}$ to inhibit its transcription, thereby decreasing HSAF production. Under iron sufficiency, the expression of Fur is low and free iron is incorporated as a co-repressor to inhibit the expression of iron-uptake genes, thereby avoiding the Fenton reaction due to excessive iron. Simultaneously, the DNA-binding affinity of Fur to $P_{HSAF}$ is weakened or even lost, which, in turn, facilitates the gene expression pattern of HSAF biosynthesis, resulting in the high production of HSAF.

Thus, the mechanism by which iron ion enhances HSAF production was explored herein, providing theoretical basis for improving HSAF production through genetic modification of strains.

## MATERIALS AND METHODS

### Bacterial strains, plasmids, and growth conditions

All bacterial strains and plasmids used in this study are listed in Table 1. *L. enzymogenes* and *E. coli* strains were cultured in LB or on LB agar plates at 28°C and 37°C, respectively. The antibiotics used were as follows: for *L. enzymogenes*, 100 µg/mL kanamycin (Km) and 150 µg/mL gentamicin (Gm); and for *E. coli*, 50 µg/mL Km, 50 µg/mL Gm, 12.5 µg/mL tetracycline, and 34 µg/mL chloramphenicol (Cm).

### Fermentation of *L. enzymogenes* OH11 and its deletion mutant strains

A few loopfuls of *strains* were inoculated in a 500-mL shake-flask with 100 mL of LB medium and then aerobically incubated at 28°C for 12 h with shaking at 180 rpm. The resulting seed culture (2.5%, vol/vol) was transferred to a 500-mL flask containing 100 mL of CDM502 with or without iron ion. The initial pH of the culture medium was adjusted to (7.0 ± 0.1) by adding 2-mol/L NaOH or 2 mol/L HCl and sterilized at 121°C for 15 min. An amount of 50 g/L aqueous solution of $FeCl_3 \cdot 6H_2O$ and $FeSO_4 \cdot 7H_2O$ was prepared as needed, filtered for sterilization, and added in the medium with a certain amount to achieve the predetermined concentration before cultivation. The fermentation flasks were incubated at 28°C with shaking at 180 rpm for 48 h to determine the optical density

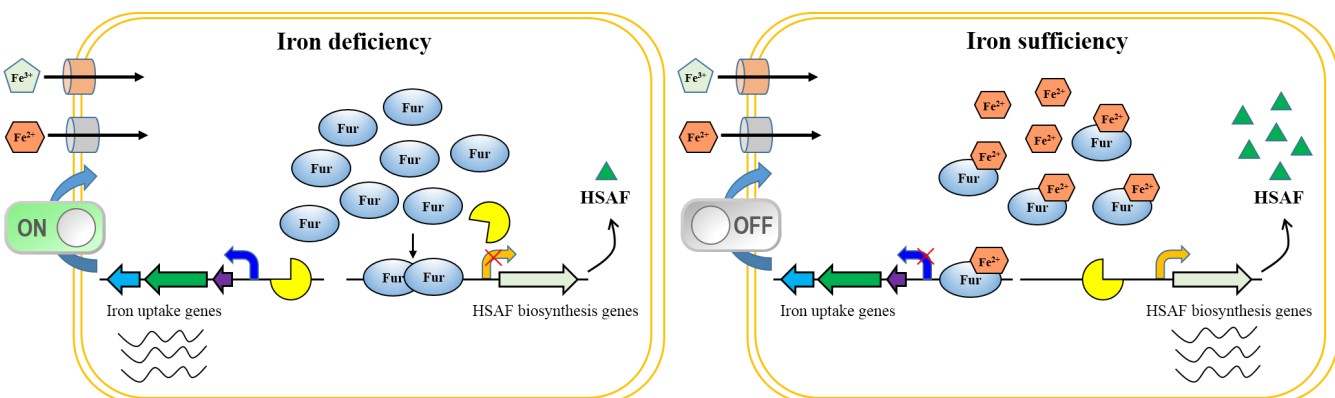

**FIG 8**  Modeling of Fur-mediated iron uptake and HSAF biosynthesis in *L. enzymogenes OH11*. Iron deficiency induces high expression of Fur but without Fur-$Fe^{2+}$ complexes as repressors. This stimulates the iron uptake process, leading to increased iron acquisition from the environment. Fur can also bind to $P_{HSAF}$ to inhibit its transcription, decreasing HSAF production. Under iron sufficiency, low Fur expression and incorporation of free iron as a co-repressor inhibits iron-uptake genes, preventing the Fenton reaction. Fur's DNA-binding affinity to $P_{HSAF}$ is weakened, leading to increased gene expression for HSAF biosynthesis and high HSAF production.

**TABLE 1** Strains and plasmids used in this study

| Strains | Characteristics[a] | Source |
|---------|-------------------|--------|
| *L. enzymogenes* OH11 | *WT* strain, Km$^R$ | Lab stock |
| Δ*bp* | In-frame deletion of *OH11GL003062*, Km$^R$ | This study |
| Δ*bfr* | In-frame deletion of *OH11GL002596*, Km$^R$ | This study |
| Δ*fitpA* | In-frame deletion of *OH11GL005044*, Km$^R$ | This study |
| Δ*Rrf2* | In-frame deletion of *OH11GL003187*, Km$^R$ | This study |
| Δ*fur* | In-frame deletion of *OH11GL004785*, Km$^R$ | This study |
| Δ*dps* | In-frame deletion of *OH11GL000618*, Km$^R$ | This study |
| Δ*fur* (*fur*) | Mutant Δ*fur* harboring plasmid pBBR-*fur*, Km$^R$ , Gm$^R$ | This study |
| *WT* (*fur*) | *WT* strain harboring plasmid pBBR-*fur*, Km$^R$ , Gm$^R$ | This study |
| E.*E. coli* DH5α | F⁻,Φ80d*lacZ*ΔM15,Δ(*lacZYA-argF*)U169, *endA1*, *recA1*, *hsdR17*(rk⁻,mk⁺), *supE44*, λ⁻*thi-1*, *gyr96*, *relA1*, *phoA* | TransGen Biotech |
| E.*E. coli* BL21 (DE3) | F⁻, *ompT*, *hdsSB*(rB⁻mB⁻), λ(DE3) | TransGen Biotech |
| XL1-Blue-MRF' Kan | Δ(*mcrA*)183, Δ(*mcrCB-hsdSMR-mrr*)l73, *endA1*, *supE44*, *thi-1*, *recA1* *gyrA96*, *relA1*, *lac*, [F' *proAB lacIqZ*ΔM15 Tn5 (Km$^R$)] | Lab stock |
| Plasmids | | |
| pEX18Gm | Suicide vector with *sacB* and Gm$^R$ | Lab stock |
| pBBR1MCS5 | Broad host range cloning vector, *lacZ*, Gm$^R$ | Lab stock |
| pET-30a (+) | Plasmid used for protein expression , Km$^R$ | Lab stock |
| pTRG | Plasmid used for protein expression in bacterial one-hybridization assay, Tet$^R$ | Lab stock |
| pBXcmT | Plasmid used for DNA cloning in bacterial one-hybridization assay, Cm$^R$ | Lab stock |
| pBBR-*fur* | pBBR1MCS5 with the coding region of *fur*, Gm$^R$ | This study |
| pET-*fur* | pET-30a with the coding region of *fur*, Km$^R$ | This study |
| pTRG-*fur* | pTRG with the coding region of *fur*, Tet$^R$ | This study |
| pBXcmT-P$_{HSAF}$ | pBXcmT with the HSAF promoter region, Cm$^R$ | This study |

[a]Km$^R$, Gm$^R$, Tet$^R$, and Cm$^R$ = resistant to kanamycin, gentamicin, tetracycline, and chloramphenicol, respectively.

at 600 nm (OD$_{600}$) and HSAF production (17). The assays were performed in triplicate. The CDM502 medium was composed of 8-g/L glucose, 1-g/L (NH$_4$)$_2$SO$_4$, 1-g/L K$_2$HPO$_4$, 0.5-g/L KH$_2$PO$_4$, and 1-g/L CaCO$_3$. The iron deficiency medium (IDM) contained CDM502, and the iron sufficiency medium (ISM) contained CDM502 + 16-mg/L FeCl$_3$.

## Construction of the deletion mutant and complement strains

Deletion mutant strains of *L. enzymogenes* OH11 were generated using a double-cross-over homologue recombination strategy, as previously described (40). Briefly, the flanking regions of each gene were amplified by PCR using the corresponding primer pairs (Table 2) and ligated into the suicide plasmid pEX18Gm. Then, these recombinant plasmids were each transformed into *WT* strain by electroporation, and the single-cross-over recombinants were picked and inoculated onto LB plates containing Km and Gm. Positive colonies were then transferred into LB without any antibiotic for 6 h and further spread on LB plates supplemented with 10% (wt/vol) sucrose and Km (25 µg/mL) for double-crossover enrichment. Significantly, 50 µL of ferric ion must be added to the sucrose plate for the knockout of *fur*. The sucrose-resistant, Km-resistant, and Gm-sensitive colonies representing double crossovers were screened and confirmed by PCR.

To construct the complemented strain of Δ*fur*, the *fur* gene was amplified by the primer pBBR-4785-F/pBBR-4785-R (Table 2) and then subcloned into the plasmid pBBR1MCS5. The resulting plasmid was introduced into Δ*fur* by electroporation as described above. The positive clone was selected on LB plates supplemented with Km (100 µg/mL) and confirmed by PCR. The complemented strain was termed Δ*fur* (*fur*). In the same way, an overexpression strain was constructed and termed *WT* (*fur*).

## HSAF extraction from fermentation and quantitative determination

Three-milliliter aliquots of fermentation samples were withdrawn from the flasks and adjusted to pH 2.5 by adding HCl. Ethyl acetate was added to the acidified broth in a 1:1

**TABLE 2** Primers used in this study

| Primer names | Sequence (5′ → 3′) |
|---|---|
| *fur-L-up* | CGACGGCCAGTGCCAAGCTTACTTGGTCAGCGCGTCGTT |
| *fur-L-do* | GGTCGCGACGCGGCCGGGGGTACGTCCCTTCGA |
| *fur-R-up* | AAGGGACGTACCCCCGGCCGCGTCGCGAC |
| *fur-R-do* | GTACCCGGGGATCCTCTAGACGTCCGGGGGGGAGCAT |
| *bp-L-up* | CGACGGCCAGTGCCAAGCTTCAGCGCAACAACGCCGG |
| *bp-L-do* | TCCAGCGGCGCTTGCGCGCGGGCCTGCTGC |
| *bp-R-up* | GCAGCAGGCCCGCGCGCAAGCGCCGCTGGAAAC |
| *bp-R-do* | GTACCCGGGGATCCTCTAGAGATGCCGGTGCGGATATTGAA |
| *bfr-L-up* | CGACGGCCAGTGCCAAGCTTCAACGACCGGCTGGTCTG |
| *bfr-L-do* | CGGTTGCGGTGGCGCGGCGCTGGTCTTCGGTTG |
| *bfr-R-up* | CCGAAGACCAGCGCCGCGCCACCGCAACCGT |
| *bfr-R-do* | GTACCCGGGGATCCTCTAGAGACGATCACCCTGGTGCT |
| *fitpA-L-up* | CGACGGCCAGTGCCAAGCTTGCGATGTCGCAGCAGGC |
| *fitpA-L-do* | CTCATCGATGCCTCCGGGGCGGCTTCCTACG |
| *fitpA-R-up* | GTAGGAAGCCGCCCCGGAGGCATCGATGAGCGC |
| *fitpA-R-do* | GTACCCGGGGATCCTCTAGATCCGGGGCTGCGGGA |
| *Rrf2-L-up* | CGACGGCCAGTGCCAAGCTTCGAGCAGCTGTGGTTGATC |
| *Rrf2-L-do* | GCGGCTGCCTACTGTCGGCGTGGGGGCGCT |
| *Rrf2-R-up* | AGCGCCCCACGCCGACAGTAGGCAGCCGCCAT |
| *Rrf2-R-do* | GTACCCGGGGATCCTCTAGATCTCGGCCAGTTCCTTGC |
| *dps-L-up* | CGACGGCCAGTGCCAAGCTTGGCCGGCGGCCAGG |
| *dps-L-do* | CGGTTCCCGGGCCGGTGGGGGAGGTGGTTCCG |
| *dps-R-up* | GAACCACCTCCCCCACCGGCCCGGGAACC |
| *dps-R-do* | GTACCCGGGGATCCTCTAGCATCAGCGCGATCAACG |
| pBBR-*4785*-F | TCGACGGTATCGATAAGCTTGCACCTGCTGCTTGCTCAT |
| pBBR-*4785*-R | GCTCTAGAACTAGTGGATCCTCAGTGGTGGTGGTGGTGGTGGCGCTTCTTGCGCACGTACA |
| pET-30a-*4785*-F | CCATGGCTGATATCGGATCCATGGAATCTCAGGACCTGCG |
| pET-30a-*4785*-R | TCGAGTGCGGCCGCAAGCTTGCGCTTCTTGCGCACGTA |
| pTRG-*4785*-F | AACCAGAGGCGGCCGGATCCATGGAATCTCAGGACCTGCG |
| pTRG-*4785*-R | ATTCTTGCGGCCGCGGATCCTCAGCGCTTCTTGCGCAC |
| pBXcmT-P$_{HSAF}$-F | CCGCTCGAGATTCCAAAGAATGATCCGCG |
| pBXcmT-P$_{HSAF}$-R | TGCTCTAGACAGCAGCGGGTGGGCGCAGT |

(vol/vol) proportion, and the mixture was shaken in a vortex mixer at 2,000 rpm for 1 min. After centrifugation, 1 mL of the solvent layer containing the HSAF was separated and ventilated to dryness in a fume hood. The HSAF crude extract was re-dissolved in 1 mL of methanol and used for high-performance liquid chromatography (Agilent 1260, USA) analysis using an InterSustainSwift C18 column (5 µm, 250 mm × 4.6 mm) with detection at 318 nm. Both pure water and acetonitrile containing 0.04% (vol/vol) trifluoracetic acid were used as the A and B mobile phases with a flow rate of 1.0 mL/min, respectively. The gradient program was as follows: 5%–25% B in 0–5 min, 25%–80% B in 5–25 min, 80%–100% B in 25–26 min, maintained to 28 min, back to 5% B at 29 min and maintained to 30 min. Finally, the production of HSAF (mg/L) was calculated from the standard curve plotted using the purified HSAF concentration and the absorption peak area (16).

## Determination of intracellular iron content

The strains of *WT*, Δ*fur*, and Δ*fur* (*fur*) were inoculated in the IDM and ISM for 24 h according to the above-mentioned method and then collected by centrifugation (8,000 rpm, 5 min). Bacterial thalli were washed twice with phosphate-buffered saline (PBS) containing 5-mM EDTA and then washed twice with PBS buffer containing no EDTA to remove residual iron ions. The collected pellets were dried overnight at 50°C. Dried pellets (0.1 g) were digested in 7 mL of HNO$_3$ at 80°C for 12 h and then diluted to

100 mL with ultrapure water. Subsequently, the total cellular iron content was measured by ICP-MS (iCAP Qc, USA).

## RNA sequencing and data analysis

The *WT* strain was cultured in IDM and ISM for 24 h according to the above fermentation method, and the cells were collected by centrifugation (10,000 rpm at 4°C for 5 min) and stored at −80°C. Total RNAs of each sample were extracted using TRIzol reagent (Invitrogen, USA), and their quality and concentration were determined by agarose gel electrophoresis and by NanoDrop, ND-1000. cDNA was synthesized using a PrimeScript reagent Kit (Takara, Japan) according to the manufacturer's instructions. High-throughput transcriptome sequencing was performed on the Illumina NovaSeq sequencing platform (NovaSeq) by Nanjing Genepioneer Bioinformatics Technology Co. Ltd. (Nanjing, Jiangsu, China). The DEGs of OH11 in the two media were analyzed using the R package DESeq2. The resulting *P* values were adjusted using the Benjamini and Hochberg approach for controlling the false discovery rate (FDR). Genes with an adjusted FDR of <5% and fold change of ≥2.0, as determined by DESeq2, were assigned as DEGs, and a set of 20 genes (10 upregulated and 10 downregulated) were randomly selected for validation with RT-qPCR. A Gene Ontology enrichment analysis of DEGs was executed using the R package clusterProfiler, and the statistical enrichments of DEGs in the Kyoto Encyclopedia of Genes and Genomes pathways were compared with the entire genome using KOBAS software.

## RNA isolation and RT-qPCR

The transcription of target genes was detected using RT-qPCR. The *WT* and Δ*fur* were cultured for 24 h in the IDM and ISM, following which the cells were collected. Total RNA was isolated using TRIzol reagent (Invitrogen) and reverse-transcribed into cDNA with the PrimeScript RT Reagent Kit with Genomic DNA Eraser (Takara) according to the manufacturer's instructions. The RT-qPCR was carried out using SYBR Premix Ex Taq II Kit (Takara) on a QuantStudio 6 Flex (Thermo Fisher Scientific, USA). Primers for the RT-qPCR are listed in Table 3. The transcription levels of selected genes were calculated using the $2^{-\Delta\Delta Ct}$ relative method with 16S rRNA serving as the endogenous reference control. The results from three replicates were averaged.

## Bioinformatics analysis of Fur

The amino acid sequences of the Fur protein in different bacteria were determined by Protein BLAST (https://blast.ncbi.nlm.nih.gov/Blast.cgi) and then aligned using ESPript version 3.0 (https://espript.ibcp.fr/ESPript/cgi-bin/ESPript.cgi) (41). The three-dimensional structure model of Fur in OH11 was prepared using the Swiss-Model Repository program (http://swissmodel.expasy.org) (42) and AlphaFold database (43) with the Fur from *Pseudomonas aeruginosa* as the reference (PDB: 3HHG), which was downloaded from PDB (https://www.rcsb.org/). Protein structural alignment of Fur between OH11 and *Pseudomonas aeruginosa* was performed by TM-align (https://zhanggroup.org/TM-align/)

**TABLE 3** Primers for RT-qPCR

| Primer names | Sequence (5′ → 3′) |
| --- | --- |
| RT-*1789*-up | CCAGGCGTCCGGTAACACCT |
| RT-*1789*-do | TGCTGGGCAGGCGGAGTCAT |
| RT-*2528*-up | CGCGCAAAACGATGAACGAG |
| RT-*2528*-do | AGGATGGTGAGCTCGCCCGA |
| RT-*3262*-up | ACGAAGACGACGCTCCTCGC |
| RT-*3262*-do | AATCGGCGGCGGCAGAACTC |
| RT-16S-up | ACATTTGATGAACGTCGGCG |
| RT-16S-do | CCACTTTCACCCGTAGGTCG |

(44). The theoretical pI and MW of Fur were predicted using the ExPASy tool (https://www.expasy.org/).

## Protein expression and purification

The native Fur protein was expressed and purified according to conventional protocols (45). First, the *fur* gene was amplified from OH11 genomic DNA and subcloned into a pET-30a (+) plasmid, following which it was transformed into *E. coli* BL21 (DE3). The transformed strain was cultivated at 37°C in LB medium to an appropriate $OD_{600}$ of 0.4 and induced with 0.2-mM isopropyl β-D-thiogalactoside at 16°C for 12 h in an incubator shaker. The cells were then collected, washed, and re-suspended in lysis buffer (15-mL PBS, pH 7.0, and 1-mM phenylmethanesulfonyl fluoride) for ultrasonication. The supernatant was incubated with Ni-NTA agarose (Sigma–Aldrich, USA) at 4°C for 2 h, washed with PBS containing 40-mM imidazole, and eluted using elution buffer containing 300-mM imidazole. Finally, the protein purity and concentration were analyzed by SDS-PAGE and Bradford protein assay kit (Bio-Rad, USA), respectively.

## EMSA

The binding of Fur to the $P_{HSAF}$ was determined by EMSA *in vitro* (46). DNA fragments corresponding to the HSAF promoter region (about 400 bp) were amplified by PCR using the primer pair pBXcmT-$P_{HSAF}$-F/pBXcmT-$P_{HSAF}$-R (Table 2). The DNA probe (20 ng) was incubated with different concentrations of His6-Fur protein in 20 µL of binding buffer [2-mM Tris-HCl (pH 7.8), 0.1-mM EDTA, 0.2-mM dithiothreitol, 4-mM KCl, 0.5-mM $MgCl_2$, 10-ng/mL bovine serum albumin, and 10% glycerol] containing 1 µg of poly(dI-dC) for 10 min at 25°C. $FeSO_4$ was added to the reaction buffer when necessary. The reaction mixtures were resolved by 5% native polyacrylamide gel at 100 V for 90 min in 0.5 × Tris-borate-EDTA buffer. Light and low temperature were avoided in this experiment. The gel was soaked in 10,000-fold-diluted SYBR Green I nucleic acid dye (Sangon Biotech, China) for 20 min and visualized using a Typhoon Scanner and Imagequant software (GE Healthcare, USA).

## Bacterial one-hybrid assay

The bacterial one-hybrid reporting system was used to confirm the physical interaction of the transcription factor with the target gene promoter *in vivo* as previously described (47). In this study, the coding regions of the *fur* (400 bp) and $P_{HSAF}$ (400 bp) were subcloned into pTRG and pBXcmT, respectively. The two recombinants were transformed into *E. coli* XL1-Blue MRF' Kan strain and spotted onto selective medium containing 5-mM 3-amino-1,2,4-triazole, 8-µg/mL streptomycin, 12.5-µg/mL tetracycline, 34-µg/mL Cm, and 30-µg/mL Km. If the positive-transformed *E. coli* strain grew well, it meant that direct physical binding occurred between fur and the $P_{HSAF}$. The co-transformant containing the vector pBX-R2031/pTRG-R3133 was used as a positive control, while the co-transformant containing the empty pTRG and pBXcmT-chiA served as a negative control. All co-transformants were grown at 28°C for 3–4 days.

## Statistical analyses

All experiments were performed in triplicate, and the data sets were analyzed by OriginPro version 8.6 and expressed as the means ± standard deviations. The significance of the treatment effects was calculated by Tukey's least significant difference ($P < 0.05$). Small letters correspond to significance values of different treatments.

## ACKNOWLEDGMENTS

We thank LetPub (https://www.letpub.com) for its linguistic assistance during the preparation of this manuscript.

This work was supported by grants from the National Natural Science Foundation of China (32102288), the Basal Research Fund of Jiangsu Province of China (CX(22)1010), and the Earmarked Fund for China Agriculture Research System (CARS-28).

We declare no conflicts of interest.

## AUTHOR AFFILIATIONS

[1]Jiangsu Key Laboratory for Food Quality and Safety, State Key Laboratory Cultivation Base of Ministry of Science and Technology, Jiangsu Academy of Agricultural Sciences, Institute of Plant Protection, Nanjing, Jiangsu, China
[2]School of Life Sciences, Jiangsu University, Zhengjiang, Jiangsu, China
[3]College of Plant Protection, Nanjing Agricultural University, Nanjing, Jiangsu, China
[4]College of Plant Protection, Hainan University, Haikou, China

## AUTHOR ORCIDs

Fengquan Liu http://orcid.org/0000-0001-9325-1500

## AUTHOR CONTRIBUTIONS

Bo Wang, Data curation, Investigation, Software, Writing – original draft | Zhizhou Xu, Investigation, Supervision | Rouxian Hou, Data curation, Investigation | Min Zhang, Formal analysis, Investigation | Xian Chen, Investigation | Youzhou Liu, Supervision | Fengquan Liu, Conceptualization, Funding acquisition, Project administration, Resources, Supervision, Writing – review and editing.

## DATA AVAILABILITY

The RNA-seq raw sequence data reported in this study has been submitted to NCBI GenBank under accession number PRJNA992215.

## ADDITIONAL FILES

The following material is available online.

### Supplemental Material

**Fig. S1-S3, Table S1 (Spectrum00617-23-s0001.docx).** RT-qPCR validation, OD600 and HSAF production, PCR and SDS-PAGE, and Primers for RT-qPCR.

### Open Peer Review

**PEER REVIEW HISTORY (review-history.pdf).** An accounting of the reviewer comments and feedback.

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
