## [Reviewer comments · Microbiology Spectrum]

Microbiology Spectrum

Iron ions regulate antifungal HSAF biosynthesis in *Lysobacter enzymogenes* by manipulating the DNA-binding affinity of the ferric uptake regulator (Fur)

Bao Tang, Bo Wang, Zhizhou Xu, Rouxian Hou, Min Zhang, Xian Chen, Youzhou Liu, and Fengquan Liu

Corresponding Author(s): Fengquan Liu, Nanjing Agricultural University/Jiangsu Academy of Agricultural Sciences

Review Timeline:

Submission Date:	February 10, 2023
Editorial Decision:	April 14, 2023
Revision Received:	June 13, 2023
Accepted:	July 5, 2023

Editor: G. Marcela Rodriguez

Reviewer(s): Disclosure of reviewer identity is with reference to reviewer comments included in decision letter(s). The following individuals involved in review of your submission have agreed to reveal their identity: Gao Chen (Reviewer #4)

Transaction Report:

DOI: <https://doi.org/10.1128/spectrum.00617-23>

April 14, 2023

Prof. Fengquan Liu
Jiangsu Academy of Agricultural Sciences
Nanjing
China

Re: Spectrum00617-23 (Iron ions regulate antifungal HSAF biosynthesis in *Lysobacter* enzymogenes by manipulating the DNA-binding affinity of the ferric uptake regulator (Fur))

Dear Prof. Fengquan Liu:

Thank you for submitting your manuscript to Microbiology Spectrum. Both reviewers agreed that the manuscript needs modifications in content and presentation to be acceptable for publication. When submitting the revised version of your paper, please provide (1) point-by-point responses to the issues raised by the reviewers as file type "Response to Reviewers," not in your cover letter, and (2) a PDF file that indicates the changes from the original submission (by highlighting or underlining the changes) as file type "Marked Up Manuscript - For Review Only". Please use this link to submit your revised manuscript - we strongly recommend that you submit your paper within the next 60 days or reach out to me. Detailed instructions on submitting your revised paper are below.

Link Not Available

Sincerely,

G. Marcela Rodriguez

Journals Department
Reviewer comments:

Reviewer #1 (Comments for the Author):

The authors studied the regulatory synthesis of HSAF (heat-stable antifungal factor) by *Lysobacter* enzymogenes, a secondary metabolite with broad-spectrum antifungal activity, but is produced in low amounts. The authors report that iron ions and the transcription factor (TF) Fur were involved in this regulatory process with Apo-Fur, in which the TF modulates gene expression in the absence of a ligand, in that case, iron. According to the authors, the improvement in HSAF production could lead to advances in the development of a biological pesticide. So far, a few transcription factors and the fermentation medium have been identified as important players in the increase of HSAF production, but not enough for large scale synthesis. The authors have evidence that the nutrient composition of the medium is important to increase HSAF production, as nutrient-rich medium the amount of HSAF synthesized is very low when compared to nutrient starved medium.

1. The authors stated in the introduction that the findings of this manuscript represent a "previously uncharacterized function of Fur in regulating secondary metabolism" (lines 101-102). This is a statement that needs further clarification for the reader: is this because Fur is modulating HSAF expression by a Apo-Fur mechanism? Or is this because Fur (and iron) are modulating HSAF expression, a secondary metabolite and that's the new finding of this paper? In both cases I'd recommend further look at the literature to check if this is really the case. (for example: doi: 10.1128/mbio.03814-21, or DOI: 10.1128/IAI.00659-16).
2. The results section starts with the investigation of the role of metals in HSAF production by *L. enzymogenes* and the effects on cell growth. Have the authors investigated any other compound that could increase HSAF production? What led you to directly test metals instead of sugars, amino acids, or other molecules that could increase this NP synthesis?
3. None of the figures had subtitles. I just found the Figure Captions (lines 651-670) and the figures, so I was not able to check, for example, what is the significance of the statistical analysis from any of the data - assuming that the letters on top of the bars and elsewhere indicate that.
4. Lines 112-113: it's known that ferrous iron is oxidized in acidic solution, but that was not proved in the experiments, as stated in this phrase. I'd suggest to merge this phrase with the following to explain why ferric iron was chosen for the subsequent experiments.
5. What is the tolerance of *L. enzymogenes* to iron? The addition of 16 mg/L could impair growth or fitness of the cell?
6. Lines 126-127: The random choice of targets for validation of the RNA-seq data is not shown. The choice took into consideration the fold-change and statistical significance? (for example, choice of up or downregulated genes with higher difference in fold-change compared to untreated conditions?)
7. Lines 139-141: it would be interesting to add the ORF of this gene that is involved in the biosynthesis of HSAF. In the introduction (lines 75-80) the authors depicted the pathway for HSAF production as a complex cascade involving several genes. Only one gene was found in the iron-stimulon data? The gene with 1.29-fold change in expression is involved exclusively in HSAF production? Or is it known to be involved in any other process in *L. enzymogenes* metabolism?
8. Lines 154-156: Where is this data?
9. Figure 3A: the OD600 measured in which time point? Was this a growth curve? If so, it would be interesting to show and interpret the whole data instead of a single point.
10. Line 162: is the unit used here correct?
11. Figure 3B: what is the conclusion about the Δfur strain producing more HSAF in the absence of iron supplementation or low amounts of iron (less or equal to 4 mg/L)?
12. Lines 173-187: is this analysis useful to support the author's experiments? This looks more like a supplementary data, or could be removed that would not impact the interpretation of the data.
13. Figure 5: is the iron concentration measured in IDM really statistically significant? How many replicates were used and which statistical test was applied here?
14. Lines 195-196: is it common to have a reduction of intracellular iron in Δfur mutants? Even though the result is consistent with previous data of the authors (Fig. 3) does it make sense?
15. Line 198, 212: replace "improvement" for "increase".
16. Line 202: replace "Fur" for "fur", as this is a reference to the gene and not the protein.
17. Line 216: Is P-HSAF the promoter region of which gene?
18. Figure 7A: is it necessary to show the amplification data?
19. Figure 7B: in line 224 is stated that the migration was at ~21 kDa, however, in the figure is shown a migration pattern above the 22 kDa band of the molecular marker. Is this a typo in the figure or in the text?
20. About the gel shift assay:
 - a. Overall the EMSA data do not show any controls. Is the promoter region of a gene related to HSAF synthesis the only target that was tested?
 - b. In the figure is not clear if there's a dose dependent response to increased Fur concentrations.
 - c. It was indicated the volume but not the concentration of Fur used in this assay.
21. I'd strongly recommend the authors to review all the citations of the paper. There are papers cited in the wrong place.
22. Line 274: strains or species?
23. Figure 8. It is shown that under iron deficient conditions HSAF production is lower than in iron sufficient conditions. In the introduction (lines 91-93) it was stated that in nutrient-rich medium, like LB, very little HSAF is synthesized. The iron levels in nutrient-rich medium are low? Or not sufficient to HSAF production?
24. Line 315: this is not a novel functional role of Fur. As acknowledged by the authors, Fur can bind in its apo or holo conformation.
25. Line 341: replace "coated" by "spread", or "streaked".
26. Line 353: "pH 2.5 by adding HCl" or "pH 2.5 with HCl".
27. Line 381: "clusterProfiler"
28. Line 392: " $2^{-\Delta\Delta Ct}$ "

Reviewer #2 (Comments for the Author):

Tang et al observed that the amendment of iron stimulated HSAF production by *Lysobacter enzymogenes*. The combination of

transcriptomics, genetics, molecular and physiological analyses elucidated the regulatory mechanism of ferric uptake regulator (Fur) on HSAF production, which has the potential to serve as an antifungal biocontrol reagent. Fur is a protein found in many bacteria that regulates the expression of genes involved in iron uptake and storage. It functions as a transcriptional repressor, meaning it binds to specific DNA sequences (called Fur boxes) in the promoter regions of target genes, preventing RNA polymerase from initiating transcription.

A lot of experiments have been performed in this study and the conclusion is solid. The finding is interesting and has novelty. However, the manuscript is poorly written, and the results and structure are not presented in a clear way for the readers to easily understand. There are many grammar mistakes, and the writing needs great improvement. Here are a few major comments.

1. it is not clear to the reviewer how the iron ions were added? Under neutral pH, ferric ion (Fe^{3+}) forms precipitates with hydroxyl group, and ferrous ion (Fe^{2+}) is unstable and will be quickly oxidized. The reviewer could not find this information in the method section.
2. L113, this statement is incorrect here. Ferrous ion (Fe^{2+}) is relatively stable under acidic conditions. Please check it up and revise.
3. L106, what does "HSAF fermentation" mean? Based on the text, HSAF is a secondary metabolite. How can it be fermented? Throughout, HSAF fermentation was mentioned in many places. However, it is not clear what it actually means. Please check and revise.
4. Fur has been intensively studied in other bacteria. It is necessary to briefly describe fur system early in the Introduction part and give the readers a basic understanding about it. L275-L289 is an introduction of Fur, and they should appear in Introduction, not in Discussion section.
5. The reviewer suggests removing the general description of transcriptome analysis (L124-141) and focusing on the discovery of the upregulation of fur genes.
6. Figure legends are too short and simple. The legends need to provide enough information for the readers to understand the figures without needing to carefully read the paper. For example in Figure 1, what the concentrations of different divalent metal ions were used in the comparison? What days were the OD600 measured?
7. Fig.2, need to directly indicate the comparison conditions for the fold change in A. Panel B is not that useful. There are a lot of non-relevant information in Fig. 2B and was intensively described in L128-L141. However, they are not so related. Instead, the genes encoding the Fur system should be pointed out and directly labeled in Fig.2A. It will be also helpful that RT-PCR can be used for the fur genes to complementarily validate the transcriptome sequencing data.

Specific comments,

L25, it might be better to add specific number or range to describe "the current production of HSAF" and describe what the requirements are?

L28, delete "identified to be".

L69, next to Streptomyces. What does it mean? Please rephrase.

L71, add the word "the" before "heat-stable".

L95, natural medium? Does it refer to rich medium? Was defined medium (such as basal salt medium) used in this study to compare with rich medium?

L110, add the word "some" before "divalent".

L113, how was ferric ion (Fe^{3+}) added and in which form? Ferric ion precipitates at neutral pH?

L117, change "acquired" to "achieved".

L126, how about showing the results of qRT-PCR in supporting information? qPCR or RT-PCR or RT-qPCR? Quantitative real-time PCR is generally abbreviated as qPCR. RT-PCR usually means reverse transcription PCR.

L144-150, OH11GLxxxxx? What are these numbers? Are they gene locus tags? Has the genome of strain OH11 been deposited to GenBank or IMG database? If so, the gene IDs need to be shown here?

L153, what are "these genes"? Need to specify.

L156, delete "somewhat".

L158, "HSAF production is determined by iron content" is not the right description.

L168, what is the difference between WT and WT (fur)

L179, replace "and" with a "-".

L271-272, what are double positive colonies?

L282, related references are needed here.

L328, N is an obsolete unit. Please change it to M (mol/L).

L328, how the fermentation condition was achieved? Were the flasks sealed and how the oxygen was removed?

L331, isn't CaCO_3 (1g/L) non-soluble in water?

L353, add (vol/vol) after 1:1.

L359-360, not clear. Do both pure water and acetonitrile contain 0.04% TFA? Or only acetonitrile? Did it run in a gradient? Need to provide details of the method or cite a reference here.

L363, how was the purified HSAF quantified? Add the information here or cite a reference.

L372-373, has the culture been passaged in the respective IDM or ISM medium several times to allow the adaptation and induction of specific genes under specific conditions? This needs to be clearly stated.

L375-376, the transcriptome sequencing and detailed data analysis protocols need to be included. Just using the company's name is not right. The methods need to contain enough information for others to repeat the experiment and judge the quality of the data.

L384, better refer it as RT-qPCR.

L406, conventional protocols, cite a reference here.

L662, replace "deepened" with "highlighted".

Staff Comments:

Preparing Revision Guidelines

Please return the manuscript within 60 days; if you cannot complete the modification within this time period, please contact me. If you do not wish to modify the manuscript and prefer to submit it to another journal, please notify me of your decision immediately so that the manuscript may be formally withdrawn from consideration by Microbiology Spectrum.

The authors studied the regulatory synthesis of HSAF (heat-stable antifungal factor) by *Lysobacter enzymogenes*, a secondary metabolite with broad-spectrum antifungal activity, but is produced in low amounts. The authors report that iron ions and the transcription factor (TF) Fur were involved in this regulatory process with Apo-Fur, in which the TF modulates gene expression in the absence of a ligand, in that case, iron. According to the authors, the improvement in HSAF production could lead to advances in the development of a biological pesticide. So far, a few transcription factors and the fermentation medium have been identified as important players in the increase of HSAF production, but not enough for large scale synthesis.

The authors have evidence that the nutrient composition of the medium is important to increase HSAF production, as nutrient-rich medium the amount of HSAF synthesized is very low when compared to nutrient starved medium.

1. The authors stated in the introduction that the findings of this manuscript represent a “previously uncharacterized function of Fur in regulating secondary metabolism” (lines 101-102). This is a statement that needs further clarification for the reader: is this because Fur is modulating HSAF expression by a Apo-Fur mechanism? Or is this because Fur (and iron) are modulating HSAF expression, a secondary metabolite and that’s the new finding of this paper? In both cases I’d recommend further look at the literature to check if this is really the case. (for example: doi: 10.1128/mbio.03814-21, or DOI: 10.1128/IAI.00659-16).
2. The results section starts with the investigation of the role of metals in HSAF production by *L. enzymogenes* and the effects on cell growth. Have the authors investigated any other compound that could increase HSAF production? What led you to directly test metals instead of sugars, amino acids, or other molecules that could increase this NP synthesis?
3. None of the figures had subtitles. I just found the Figure Captions (lines 651-670) and the figures, so I was not able to check, for example, what is the significance of the statistical analysis from any of the data – assuming that the letters on top of the bars and elsewhere indicate that.
4. Lines 112-113: it’s known that ferrous iron is oxidized in acidic solution, but that was not proved in the experiments, as stated in this phrase. I’d suggest to merge this phrase with the following to explain why ferric iron was chosen for the subsequent experiments.
5. What is the tolerance of *L. enzymogenes* to iron? The addition of 16 mg/L could impair growth or fitness of the cell?
6. Lines 126-127: The random choice of targets for validation of the RNA-seq data is not shown. The choice took into consideration the fold-change and statistical significance? (for example, choice of up or downregulated genes with higher difference in fold-change compared to untreated conditions?)
7. Lines 139-141: it would be interesting to add the ORF of this gene that is involved in the biosynthesis of HSAF. In the introduction (lines 75-80) the authors depicted the pathway

for HSAF production as a complex cascade involving several genes. Only one gene was found in the iron-stimulon data? The gene with 1.29-fold change in expression is involved exclusively in HSAF production? Or is it known to be involved in any other process in *L. enzymogenes* metabolism?

8. Lines 154-156: Where is this data?
9. Figure 3A: the OD₆₀₀ measured in which time point? Was this a growth curve? If so, it would be interesting to show and interpret the whole data instead of a single point.
10. Line 162: is the unit used here correct?
11. Figure 3B: what is the conclusion about the Δfur strain producing more HSAF in the absence of iron supplementation or low amounts of iron (less or equal to 4 mg/L)?
12. Lines 173-187: is this analysis useful to support the author's experiments? This looks more like a supplementary data, or could be removed that would not impact the interpretation of the data.
13. Figure 5: is the iron concentration measured in IDM really statistically significant? How many replicates were used and which statistical test was applied here?
14. Lines 195-196: is it common to have a reduction of intracellular iron in Δfur mutants? Even though the result is consistent with previous data of the authors (Fig. 3) does it make sense?
15. Line 198, 212: replace "improvement" for "increase".
16. Line 202: replace "Fur" for "*fur*", as this is a reference to the gene and not the protein.
17. Line 216: Is P_{HSAF} the promoter region of which gene?
18. Figure 7A: is it necessary to show the amplification data?
19. Figure 7B: in line 224 is stated that the migration was at ~21 kDa, however, in the figure is shown a migration pattern above the 22 kDa band of the molecular marker. Is this a typo in the figure or in the text?
20. About the gel shift assay:
 - a. Overall the EMSA data do not show any controls. Is the promoter region of a gene related to HSAF synthesis the only target that was tested?
 - b. In the figure is not clear if there's a dose dependent response to increased Fur concentrations.
 - c. It was indicated the volume but not the concentration of Fur used in this assay.
21. I'd strongly recommend the authors to review all the citations of the paper. There are papers cited in the wrong place.
22. Line 274: strains or species?
23. Figure 8. It is shown that under iron deficient conditions HSAF production is lower than in iron sufficient conditions. In the introduction (lines 91-93) it was stated that in nutrient-rich medium, like LB, very little HSAF is synthesized. The iron levels in nutrient-rich medium are low? Or not sufficient to HSAF production?
24. Line 315: this is not a novel functional role of Fur. As acknowledged by the authors, Fur can bind in its apo or holo conformation.
25. Line 341: replace "coated" by "spread", or "streaked".
26. Line 353: "pH 2.5 by adding HCl" or "pH 2.5 with HCl".
27. Line 381: "clusterProfiler"
28. Line 392: " $2^{-\Delta\Delta Ct}$ "

Dear G. Marcela Rodriguez, Editor

Thank you very much for your response and the reviewers' comments regarding our manuscript entitled "Iron ions regulate antifungal HSAF biosynthesis in *Lysobacter enzymogenes* by manipulating the DNA-binding affinity of the ferric uptake regulator (Fur)" (Paper #Spectrum00617-23). These comments were all valuable and helped us revise and improve our paper, and they will guide us in our future research. Revisions in the text are marked with red font, and our point-by-point responses to specific comments are listed below. We hope that the revised manuscript satisfies both you and the reviewers.

Yours sincerely,

Fengquan Liu

Responding to Reviewers' comments:

Reviewer #1 (Comments for the Author):

The authors studied the regulatory synthesis of HSAF (heat-stable antifungal factor) by *Lysobacter enzymogenes*, a secondary metabolite with broad-spectrum antifungal activity, but is produced in low amounts. The authors report that iron ions and the transcription factor (TF) Fur were involved in this regulatory process with Apo-Fur, in which the TF modulates gene expression in the absence of a ligand, in that case, iron. According to the authors, the improvement in HSAF production could lead to advances in the development of a biological pesticide. So far, a few transcription factors and the fermentation medium have been identified as important players in the increase of HSAF production, but not enough for large scale synthesis.

The authors have evidence that the nutrient composition of the medium is important to increase HSAF production, as nutrient-rich medium the amount of HSAF synthesized is very low when compared to nutrient starved medium.

1. The authors stated in the introduction that the findings of this manuscript represent a "previously uncharacterized function of Fur in regulating secondary metabolism" (lines 101-102). This is a statement that needs further clarification for the reader: is this because Fur is modulating HSAF expression by a Apo-Fur mechanism? Or is this because Fur (and iron) are modulating HSAF expression, a secondary metabolite and that's the new finding of this paper? In both cases I'd recommend further look at the literature to check if this is really the case. (for example: doi: 10.1128/mbio.03814-21, or DOI: 10.1128/IAI.00659-16).

Reply and revision: Thank you for this constructive comment. Depending on the two literatures you provided, we have changed our previous statement to “In a word, we report the function of Fur in regulating secondary metabolism in *Lysobacter* for the first time”. Thank you so much.

2. The results section starts with the investigation of the role of metals in HSAF production by *L. enzymogenes* and the effects on cell growth. Have the authors investigated any other compound that could increase HSAF production? What led you to directly test metals instead of sugars, amino acids, or other molecules that could increase this NP synthesis?

Reply and revision: Thank you for your thoughtful suggestion. Actually, we have investigated the effects of carbon sources, nitrogen sources, and inorganic salts on HSAF production in previous study (*Letters in Applied Microbiology*, 2018, 66 (5):439-446). In order to further increase the production of HSAF, the effects of different metal ions on the fermentation were investigated in this study. We have added the information in the revised manuscript.

3. None of the figures had subtitles. I just found the Figure Captions (lines 651-670) and the figures, so I was not able to check, for example, what is the significance of the statistical analysis from any of the data - assuming that the letters on top of the bars and elsewhere indicate that.

Reply and revision: We apologize for the confusion. In the newly submitted

manuscript, we have added subtitles to the figures and enriched the content of the captions, especially regarding explanations of the data analysis.

4. Lines 112-113: it's known that ferrous iron is oxidized in acidic solution, but that was not proved in the experiments, as stated in this phrase. I'd suggest to merge this phrase with the following to explain why ferric iron was chosen for the subsequent experiments.

Reply and revision: Thank you for this comment. As suggested, we have changed those sentences to “Considering the unstable nature of Fe^{2+} , it was rapidly oxidized under aerobic, moderate pH conditions, so Fe^{3+} was selected as the suitable metal ion for subsequent experiments” in resubmitted manuscript.

5. What is the tolerance of *L. enzymogenes* to iron? The addition of 16 mg/L could impair growth or fitness of the cell?

Reply and revision: As shown in Fig. 1B, the addition of different concentrations of iron ions resulted in better growth compared to the no-iron condition. However, increasing the iron concentration from 4 mg/L to 32 mg/L did not impact the growth of the bacteria. This suggests that *L. enzymogenes* has a strong tolerance to iron and that the addition of 16 mg/L did not adversely affect cell growth or fitness.

Fig. 1 Effects of metal ion types (A) and concentrations (B) on the fermentation.

6. Lines 126-127: The random choice of targets for validation of the RNA-seq data is not shown. The choice took into consideration the fold-change and statistical significance? (for example, choice of up or downregulated genes with higher difference in fold-change compared to untreated conditions?)

Reply and revision: As depicted in Fig. S1, we have added the validation results of RT-qPCR. Except for the HSAF biosynthesis gene (*OH11GL005113*), the selection of other genes mainly considered different fold-change. Thank you for your comment.

Fig. S1 RT-qPCR validation of RNA-seq data. The X axis shows the selected genes, and the Y axis shows the relative expression level of each gene.

7. Lines 139-141: it would be interesting to add the ORF of this gene that is involved in the biosynthesis of HSAF. In the introduction (lines 75-80) the authors depicted the pathway for HSAF production as a complex cascade involving several genes. Only one gene was found in the iron-stimulon data? The gene with 1.29-fold change in expression is involved exclusively in HSAF production? Or is it known to be involved in any other process in *L. enzymogenes* metabolism?

Reply and revision: Thank you for your comment. Map of the HSAF gene cluster was shown in Fig. 9 (*Journal of the American Chemical Society*, 2011, 4(133):643-645). The synthesis of HSAF mainly involved 10 genes, of which ORF6 (marked in red) was the core synthetic gene, usually referring to the HSAF synthesis gene. So ORF6 was selected for research, which has also been reported in other studies (*Applied and Environmental Microbiology*, 2017, 7(83): e03397). Besides HSAF, the HSAF gene cluster can also synthesize other structural analogues, such as a very small amount of alteramide B (ATB) in the wild-type strain (*Bioresource Technology*, 2019, 273: 196-202).

Fig. 9 Map of the HSAF gene cluster (*Journal of the American Chemical Society*, 2011, 4(133):643-645).

8. Lines 154-156: Where is this data?

Reply and revision: Your suggestion is very good. We have supplemented “Fig. S2 OD₆₀₀ and HSAF production in wild-type and mutant strain cultured in the IDM” in the “Supplementary materials”. Thank you.

9. Figure 3A: the OD₆₀₀ measured in which time point? Was this a growth curve? If so, it would be interesting to show and interpret the whole data instead of a single point.

Reply and revision: The OD₆₀₀ in the fermentation broth was measured at the 48-hour endpoint of fermentation. Fig. 3A did not show a growth curve. We have supplemented this data in the “Materials and Methods”. Thank you.

10. Line 162: is the unit used here correct?

Reply and revision: Sorry for this mistake, the unit should be mg/L. We have corrected it. Thank you.

11. Figure 3B: what is the conclusion about the Δfur strain producing more HSAF in the absence of iron supplementation or low amounts of iron (less or equal to 4 mg/L)?

Reply and revision: The result indicated that the transcription factor of Fur has a negative impact on the HSAF biosynthesis under low-iron conditions. The conclusion can be found in the “Discussion”. Thank you.

12. Lines 173-187: is this analysis useful to support the author's experiments? This looks more like a supplementary data, or could be removed that would not impact the interpretation of the data.

Reply and revision: Firstly, although Fur is commonly present in microorganisms, it has not been identified in *Lysobacter*. Therefore, it is very necessary to understand the basic properties of Fur in this study. Then, we found that Fur contained one DNA-binding domain and two metal-binding sites, which could support the subsequent results of “Fur can directly interact with P_{HSAF}, and the specific interactions were inhibited by iron ion”. Therefore, we think this analysis was useful and could not be removed. Thank you.

13. Figure 5: is the iron concentration measured really statistically significant? How many replicates were used and which statistical test was applied here?

Reply and revision: Many chemical reagents contained trace amounts of iron, even at analytical purity levels (>99%). The determination of intracellular iron content was performed in triplicate and the significance of the treatment effects was determined by Tukey's LSD, $p < 0.05$. So We think that it is reasonable to detect iron within cells when cultured in IDM, and the measurement of intracellular iron ions is statistically significant. Thank you.

14. Lines 195-196: is it common to have a reduction of intracellular iron in Δfur mutants? Even though the result is consistent with previous data of the authors (Fig. 3) does it make sense?

Reply and revision: Indeed, it is a common phenomenon, which has been reported in other studies (*Applied and Environmental Microbiology*, 2015, 23(81):8044-8053). This statement of “This result is consistent with Fig. 3A.” doesn't make sense, we have deleted the sentence. Thank you.

15. Line 198, 212: replace "improvement" for "increase".

Reply and revision: Thank you for this comment. As suggested, we have replaced "improvement" for "increase" in this resubmitted manuscript.

16. Line 202: replace "Fur" for "fur", as this is a reference to the gene and not the protein.

Reply and revision: It was our negligence. We have replaced "Fur" for "fur. Thank you.

17. Line 216: Is P^{-HSAF} the promoter region of which gene?

Reply and revision: P_{HSAF} is the promoter of the HSAF biosynthesis gene (*OHI1GL005113*). We have described its full name when P_{HSAF} first appeared.

18. Figure 7A: is it necessary to show the amplification data?

Reply and revision: Thank you for this comment. We have removed Fig. 7A and 7B to the “Supplementary materials” as Fig. S3A and Fig. S3B.

19. Figure 7B: in line 224 is stated that the migration was at ~21 kDa, however, in the figure is shown a migration pattern above the 22 kDa band of the molecular marker. Is this a typo in the figure or in the text?

Reply and revision: We feel sorry for this mistake. The molecular weight of Fur and His tag fusion protein was 22.36 kDa, we have corrected this information in the resubmitted manuscript. Thank you.

Fig. S3 (A) PCR result of P_{HSAF} ; (B) SDS-PAGE analysis of purified His-tagged Fur.

20. About the gel shift assay:

a. Overall the EMSA data do not show any controls. Is the promoter region of a gene related to HSAF synthesis the only target that was tested?

b. In the figure is not clear if there's a dose dependent response to increased Fur

concentrations.

c. It was indicated the volume but not the concentration of Fur used in this assay.

Reply and revision: We appreciate your comments. As suggested, we reorganized Fig. 7 in resubmitted manuscript. Now, we use the promoter of *OHI1GL003062* as a negative control of EMSA analysis. As shown in Fig. 7B, Fur cannot be directly bound to $P_{OHI1GL003062}$, which means that the binding between Fur to P_{HSAF} is specific. In Fig. 7A, we relabelled the concentration of Fur, now we can see that the Fur- P_{HSAF} complex could be stronger by the addition of Fur protein. However, the shift induced by the formation of the complex of the Fur-His protein with P_{HSAF} was inhibited by the addition of $FeSO_4$: as the concentration of $FeSO_4$ increased from 0 to 100 nM, the signals of the shifted binding complex bands became increasingly weaker.

Fig. 7 Iron ions inhibit the binding between Fur and P_{HSAF} . (A) Fur binds to P_{HSAF} *in vitro*. (B) Gel shift assay showing that Fur does not directly bind the *OHI1GL003062*

promoter region. (C) The direct physical interaction between Fur and P_{HSAF} was detected in *Escherichia coli* under different iron ions growth condition.

21. I'd strongly recommend the authors to review all the citations of the paper. There are papers cited in the wrong place.

Reply and revision: Sorry for this mistake, in this resubmitted manuscript we have corrected them all. We have revised reference 19, and deleted reference 22, 26, and 27. Thank you.

22. Line 274: strains or species?

Reply and revision: Thank you for this comment. We have changed strains to species.

23. Figure 8. It is shown that under iron deficient conditions HSAF production is lower than in iron sufficient conditions. In the introduction (lines 91-93) it was stated that in nutrient-rich medium, like LB, very little HSAF is synthesized. The iron levels in nutrient-rich medium are low? Or not sufficient to HSAF production?

Reply and revision: Thank you for your comment. We think that these descriptions are not contradictory. LB is nutrient-rich medium, but its specific composition is unknown as it is a natural medium. Very little HSAF is produced in LB medium, but it cannot be determined which component affects HSAF production. The results of Figure 8 occurred in the chemically defined medium, the addition of iron ions can

promote the synthesis of HSAF. Therefore, iron ions are a factor that affects the production of HSAF, but not the only factor.

24. Line 315: this is not a novel functional role of Fur. As acknowledged by the authors, Fur can bind in its apo or holo conformation.

Reply and revision: Thank you for this comment. We have reorganized this sentence to “Thus, the mechanism by which iron ion enhances HSAF production was explored herein, providing theoretical basis improving HSAF production through genetic modification of strains”.

25. Line 341: replace "coated" by "spread", or "streaked".

Reply and revision: Thank you for this comment. We have replaced "coated" by "spread".

26. Line 353: "pH 2.5 by adding HCl" or "pH 2.5 with HCl".

Reply and revision: Thank you for this comment. We have changed "pH 2.5 by HCl" to "pH 2.5 by adding HCl".

27. Line 381: "clusterProfiler"

Reply and revision: Thank you for pointing our mistake. We have corrected this writing.

28. Line 392: "2- $\Delta\Delta C_t$ "

Reply and revision: Thank you for pointing our mistake. We have corrected this writing.

Reviewer #2 (Comments for the Author):

Tang et al observed that the amendment of iron stimulated HSAF production by *Lysobacter enzymogenes*. The combination of transcriptomics, genetics, molecular and physiological analyses elucidated the regulatory mechanism of ferric uptake regulator (Fur) on HSAF production, which has the potential to serve as an antifungal biocontrol reagent. Fur is a protein found in many bacteria that regulates the expression of genes involved in iron uptake and storage. It functions as a transcriptional repressor, meaning it binds to specific DNA sequences (called Fur boxes) in the promoter regions of target genes, preventing RNA polymerase from initiating transcription.

A lot of experiments have been performed in this study and the conclusion is solid. The finding is interesting and has novelty. However, the manuscript is poorly written, and the results and structure are not presented in a clear way for the readers to easily understand. There are many grammar mistakes, and the writing needs great improvement. Here are a few major comments.

1. it is not clear to the reviewer how the iron ions were added? Under neutral pH, ferric ion (Fe^{3+}) forms precipitates with hydroxyl group, and ferrous ion (Fe^{2+}) is

unstable and will be quickly oxidized. The reviewer could not find this information in the method section.

Reply and revision: 50 g/L aqueous solution of $\text{FeCl}_3 \cdot 6\text{H}_2\text{O}$ and $\text{FeSO}_4 \cdot 7\text{H}_2\text{O}$ was prepared as needed, filtered for sterilization, and added in the medium with a certain amount to achieve the predetermined concentration before cultivation. We have supplemented this information in the material method. Thank you. (*Lines 328-330*)

Under aerobic, moderate pH conditions ferrous iron is oxidized spontaneously to the ferric (Fe^{3+}) form and is hydrolyzed abiotically to insoluble ferric hydroxide ($\text{Fe}(\text{OH})_3$). However, the spontaneous process is very slow (*Environmental Sustainability*, 2018, 3(1):221-231). In addition, the fermentation process would produce hydrogen ions and caused a decrease in pH of the fermentation broth, preventing this spontaneous process.

2. L113, this statement is incorrect here. Ferrous ion (Fe^{2+}) is relatively stable under acidic conditions. Please check it up and revise.

Reply and revision: This is our mistake. Thank you for your correction. We have corrected this statement to “Considering the unstable nature of Fe^{2+} , it was rapidly oxidized under aerobic, moderate pH conditions”.

3. L106, what does "HSAF fermentation" mean? Based on the text, HSAF is a secondary metabolite. How can it be fermented? Throughout, HSAF fermentation was

mentioned in many places. However, it is not clear what it actually means. Please check and revise.

Reply and revision: In this study, HSAF fermentation was actually equivalent to the fermentation of *L. enzymogenes* OH11. To emphasize the synthesis process of the metabolite HSAF, HSAF fermentation was colloquially used. This use has also occurred in other studies, such as succinic acid fermentation (*Bioresource Technology*, 2021, 342:126014). We have made modifications to make it easier to accept. Thank you.

4. Fur has been intensively studied in other bacteria. It is necessary to briefly describe fur system early in the Introduction part and give the readers a basic understanding about it. L275-L289 is an introduction of Fur, and they should appear in Introduction, not in Discussion section.

Reply and revision: Thank you for your suggestion. When we started writing the manuscript, we had the same idea as you. However, it was illogical to describe the transcription factor of Fur in the “Introduction” after careful consideration. Because the regulation of HSAF biosynthesis by Fur was discovered through extensive experiments. We didn't know at first. So the description of Fur should not be included in the “Introduction”.

5. The reviewer suggests removing the general description of transcriptome analysis (L124-141) and focusing on the discovery of the upregulation of fur genes.

Reply and revision: After fully considering your suggestion, we have simplified the description and focused on the discovery of Fur. Thank you for your comment.

6. Figure legends are too short and simple. The legends need to provide enough information for the readers to understand the figures without needing to carefully read the paper. For example in Figure 1, what the concentrations of different divalent metal ions were used in the comparison? What days were the OD₆₀₀ measured?

Reply and revision: As suggestion, we have added subtitles and enriched the content of the captions to make it easier for readers to understand the figures. The OD₆₀₀ in the fermentation broth was measured at the 48-hour endpoint of fermentation. We have supplemented this data in the “Materials and Methods”. Thank you.

7. Fig. 2, need to directly indicate the comparison conditions for the fold change in A. Panel B is not that useful. There are a lot of non-relevant information in Fig. 2B and was intensively described in L128-L141. However, they are not so related. Instead, the genes encoding the Fur system should be pointed out and directly labeled in Fig. 2A. It will be also helpful that RT-PCR can be used for the fur genes to complementarily validate the transcriptome sequencing data.

Reply and revision: We believe that the number of upregulated and downregulated genes shown in Fig. 2A was sufficient, and similar results have also occurred in other studies (*International Journal of Molecular Sciences*, 2018, 7(19): 1841). Fig. 2B shows COG annotation of the DEGs, and we think that this information is very

meaningful to search for the iron-response genes that regulate HSAF synthesis. In fact, six candidate genes were obtained through the COG analysis. We have deleted the lengthy description. Thank you. The validation results of RT-qPCR have been added and shown in Fig. S1.

Specific comments,

L25, it might be better to add specific number or range to describe "the current production of HSAF" and describe what the requirements are?

Reply and revision: The current production of HSAF is 440.26 ± 16.14 mg/L in a natural medium, and the requirements for large-scale production have no unified standard, usually at least 1 g/L. We have added the specific number and described the requirements in the "Introduction", not in the "abstract". Thank you for your suggestion.

L28, delete "identified to be".

Reply and revision: We have deleted "identified to be". Thank you.

L69, next to *Streptomyces*. What does it mean? Please rephrase.

Reply and revision: Sorry, this statement is not accurate. We have deleted "next to *Streptomyces*". Thank you very much.

L71, add the word "the" before "heat-stable".

Reply and revision: We have added "the" before "heat-stable antifungal factor".

Thank you.

L95, natural medium? Does it refer to rich medium? Was defined medium (such as basal salt medium) used in this study to compare with rich medium?

Reply and revision: Natural medium is a natural organic substance that contains unclear or unstable chemical components. It does not refer to rich medium. Natural medium is relative to chemically defined medium, which contains precisely known components and amounts. Thank you.

L110, add the word "some" before "divalent".

Reply and revision: We have added "some" before "divalent". Thank you. (*Line 114*)

L117, change "acquired" to "achieved".

Reply and revision: We have replaced "acquired" with "achieved" as suggested.

Thank you.

L126, how about showing the results of qRT-PCR in supporting information? qPCR or RT-PCR or RT-qPCR? Quantitative real-time PCR is generally abbreviated as qPCR. RT-PCR usually means reverse transcription PCR.

Reply and revision: The most accurate expression is “reverse transcription quantitative polymerase chain reaction (RT-qPCR)”, we have modified them. We have added the validation results of RT-qPCR, as shown in Fig. S1. Thank you.

L144-150, OH11GLxxxxx? What are these numbers? Are they gene locus tags? Has the genome of strain OH11 been deposited to GenBank or IMG database? If so, the gene IDs need to be shown here?

Reply and revision: The genome of strain OH11 has been fully sequenced, but has not been uploaded to GenBank or IMG database. These numbers are the numbers of genes in the genome. Thank you.

L153, what are "these genes"? Need to specify.

Reply and revision: We have modified "we knocked out these genes" to “six gene (*bp*, *bfr*, *fitpA*, *rrf2*, *fur*, and *dps*) deletion mutants were constructed”. Thank you for your suggestion.

L156, delete "somewhat".

Reply and revision: We have deleted "somewhat". Thank you.

L158, "HSAF production is determined by iron content" is not the right description.

Reply and revision: The suggestion was valuable. Thank you. We have revised this sentence to “However, there was a significant difference in HSAF production between Δfur and *WT* under different concentrations of iron ions”.

L168, what is the difference between *WT* and *WT (fur)*

Reply and revision: *WT* refers to the wild-type strain, and *WT (fur)* was the wild-type strain containing the recombinant plasmid pBBR-*fur*, as listed in Table 1. Thank you.

L179, replace "and" with a "-".

Reply and revision: We have replaced "and" with a "-". Thank you.

L271-272, what are double positive colonies?

Reply and revision: We apologized for this inaccurate statement. We have modified “double positive colonies” to “double-crossover recombinants”. Thank you.

L328, N is an obsolete unit. Please change it to M (mol/L).

Reply and revision: We have changed “N” to “mol/L”. Thank you for this minder.

L328, how the fermentation condition was achieved? Were the flasks sealed and how the oxygen was removed?

Reply and revision: The fermentation conditions affecting the production of HSAF were optimized in our previous study (*BMC Biotechnology*, 2018, 1(18)), we have

added the reference. The strain of OH11 is an aerobic microorganism, and the flask was wrapped with a breathable film. Thank you.

L331, isn't CaCO₃ (1g/L) non-soluble in water?

Reply and revision: Yes, CaCO₃ is basically insoluble in water. CaCO₃ is usually added to the fermentation medium to buffer pH to avoid acidic environments, and can also release calcium ions for bacterial utilization. Thank you.

L353, add (vol/vol) after 1:1.

Reply and revision: We have added them, thank you.

L359-360, not clear. Do both pure water and acetonitrile contain 0.04% TFA? Or only acetonitrile? Did it run in a gradient? Need to provide details of the method or cite a reference here.

Reply and revision: Thank you for your suggestion. Both pure water and acetonitrile contain 0.04% TFA, we have revised and provided details of the gradient program.

L363, how was the purified HSAF quantified? Add the information here or cite a reference.

Reply and revision: High-purity HSAF was acquired by the preparative HPLC and the purity of HSAF was determined to be 96.54% by HPLC in our previous study

(*Letters in Applied Microbiology*, 2018, 66 (5):439-446). We have cited the reference.

Thank you very much.

L372-373, the culture has been passaged in the respective IDM or ISM medium several times to allow the adaptation and induction of specific genes under specific conditions? This needs to be clearly stated.

Reply and revision: In fact, the *WT* strain was inoculated in a 500-mL shake-flask and then aerobically incubated at 28°C for 12 h with shaking at 180 rpm. The seed culture (2.5%, v/v) was transferred to fresh IDM and ISM. The culture was the same as the above fermentation method. We have revised this statement. Thank you.

L375-376, the transcriptome sequencing and detailed data analysis protocols need to be included. Just using the company's name is not right. The methods need to contain enough information for others to repeat the experiment and judge the quality of the data.

Reply and revision: As you suggest, we have added the detailed transcriptome sequencing steps. Thank you.

L384, better refer it as RT-qPCR.

Reply and revision: As suggested, we have revised “quantitative real-time polymerase chain reaction (qRT-PCR)” to “reverse transcription quantitative polymerase chain reaction (RT-qPCR)” in the revised manuscript. Thank you.

L406, conventional protocols, cite a reference here.

Reply and revision: We have cited a reference. Thank you.

L662, replace "deepened" with "highlighted".

Reply and revision: We have replaced the word. Thank you.

June 27, 2023

Prof. Fengquan Liu
Nanjing Agricultural University/Jiangsu Academy of Agricultural Sciences
College of Plant Protection, Nanjing Agricultural University, Nanjing 210095, China /Key Laboratory of Integrated Management of Crop Diseases and Pests (Nanjing Agricultural University), Ministry of
Institute of Plant Protection, Jiangsu Academy of Agricultural Sciences, Nanjing 210014, China
nanjing, China 210014
China

Re: Spectrum00617-23R1 (Iron ions regulate antifungal HSAF biosynthesis in *Lysobacter enzymogenes* by manipulating the DNA-binding affinity of the ferric uptake regulator (Fur))

Dear Prof. Fengquan Liu:

Your manuscript has been accepted, and I am forwarding it to the ASM Journals Department for publication. You will be notified when your proofs are ready to be viewed. However, please note that in the present form the article does not comply with ASM's Data Policy (<https://journals.asm.org/open-data-policy>). You should provide a "Data Availability" paragraph at the end of the Materials and Methods section that includes data description, name of the repository of sequence data and DOIs or accession numbers. We cannot proceed to publication without this missing information.

Sincerely,

G. Marcela Rodriguez
Editor, Microbiology Spectrum

Journals Department
In their reviewed manuscript Tang et al. provide further data to corroborate the questions raised by the reviewer and support the findings of the work. Corrected and/or modifications in the data and the text were also made that, in my view, clarify and help the reader have an in-depth comprehension of the authors' goals and data interpretation. I believe that this updated version is improved and provide new knowledge about the biology of *Lysobacter enzymogenes* and HSAF production, as well as the regulatory mechanism employed by Fur in this scenario.